# Engineering indel and substitution variants of diverse and ancient enzymes using Graphical Representation of Ancestral Sequence Predictions (GRASP)

**Gabriel Foley**[1], **Ariane Mora**[1☯], **Connie M. Ross**[1☯], **Scott Bottoms**[2☯], **Leander Sützl**[3☯], **Marnie L. Lamprecht**[1], **Julian Zaugg**[1], **Alexandra Essebier**[1], **Brad Balderson**[1], **Rhys Newell**[1], **Raine E. S. Thomson**[1], **Bostjan Kobe**[1,4], **Ross T. Barnard**[1], **Luke Guddat**[1], **Gerhard Schenk**[1,5], **Jörg Carsten**[6], **Yosephine Gumulya**[1], **Burkhard Rost**[7], **Dietmar Haltrich**[3], **Volker Sieber**[1,2,6], **Elizabeth M. J. Gillam**[1]*, **Mikael Bodén**[1]*

**1** School of Chemistry and Molecular Biosciences, The University of Queensland, Brisbane, Australia, **2** Campus Straubing for Biotechnology and Sustainability, Technische Universität München, Straubing, Germany, **3** Institut für Lebensmitteltechnologie, Universität für Bodenkultur Wien, Vienna, Austria, **4** Institute for Molecular Bioscience and Australian Infectious Diseases Research Centre, The University of Queensland, Brisbane, Australia, **5** Sustainable Minerals Institute, The University of Queensland, Brisbane, Australia, **6** Zentralinstitut für Katalyseforschung, Technische Universität München, Munich, Germany, **7** Fakultät für Informatik, Technische Universität München, Munich, Germany

☯ These authors contributed equally to this work.
* m.boden@uq.edu.au (MB); e.gillam@uq.edu.au (EMJG)

**Data Availability Statement:** Data sets for CYP2U, DHAD, KARI, and GDH-GOx reconstructions are available from https://github.com/bodenlab/

## Abstract

Ancestral sequence reconstruction is a technique that is gaining widespread use in molecular evolution studies and protein engineering. Accurate reconstruction requires the ability to handle appropriately large numbers of sequences, as well as insertion and deletion (indel) events, but available approaches exhibit limitations. To address these limitations, we developed Graphical Representation of Ancestral Sequence Predictions (GRASP), which efficiently implements maximum likelihood methods to enable the inference of ancestors of families with more than 10,000 members. GRASP implements partial order graphs (POGs) to represent and infer insertion and deletion events across ancestors, enabling the identification of building blocks for protein engineering.

To validate the capacity to engineer novel proteins from realistic data, we predicted ancestor sequences across three distinct enzyme families: glucose-methanol-choline (GMC) oxidoreductases, cytochromes P450, and dihydroxy/sugar acid dehydratases (DHAD). All tested ancestors demonstrated enzymatic activity. Our study demonstrates the ability of GRASP (1) to support large data sets over 10,000 sequences and (2) to employ insertions and deletions to identify building blocks for engineering biologically active ancestors, by exploring variation over evolutionary time.

GRASP-resources. All different data sets within each family are included, representing a range of different sequence identities and sizes (S17 Fig). Included are the original FASTA files, phylogenetic trees, sets of ancestors, and phylogenetic trees with labelled ancestral nodes. The full set of ancestors from each joint reconstruction is provided, except for GDH-GOx, where the single nominated marginal reconstruction at N320 is provided.

**Funding:** This work has been supported by the Australian Research Council (ARC; arc.gov.au) Discovery Project grants 210101802 to GS, LG and MB, 160100865 to MB, EG, BK and BR and 120101772 to MB and EG. BK is an ARC Laureate Fellow (FL180100109). The funders had no role in study design, data collection and analysis, decision to publish, or preparation of the manuscript.

**Competing interests:** The authors have declared that no competing interests exist.

## Author summary

Massive sequencing projects expose the extent of natural, genetic diversity. Here, we describe a method with capacity to perform ancestor sequence reconstruction from data sets in excess of 10,000 sequences, poised to recover *ancestral* diversity, including the evolutionary events that determine present-time biological function and structure.

We introduce a novel strategy for suggesting "insertion-deletion variants" that are distinct from, but can be explored alongside, substitution variants for creating ancestral libraries. We demonstrate how insertions and deletions can be used as building blocks to form "hybrid ancestors"; based on this strategy, we synthesise ancestor variants, with varying enzymatic activities, for wide-ranging applications in the biotechnology sector.

This is a *PLOS Computational Biology* Methods paper.

## Introduction

Sequencing technology is driving the identification of the *extant* (modern) portion of the universe of biological sequences [1–3]. With this increased coverage of natural diversity we are now better placed than ever before to leverage ancestral sequence reconstruction (ASR) to recover the *ancestral* portion and trace the evolutionary events that have determined biological function and structure [4, 5]. This is especially useful for protein engineering; the evolutionary record reveals essential cues for the discovery and engineering of new enzymes. The resurrection of ancestral enzymes can be used to generate enzymes with novel properties that can be exploited in biocatalysis [6–10].

The ability to perform ASR on large-scale data has been limited by the available methodology and accompanying technology. A recent review highlighted 12 studies from the past decade that each sought to evaluate sources of ambiguity in ancestral inferences [11]. Data set sizes within these studies ranged from 21 to 456 sequences, with an average of 168 sequences. Current methods for performing ASR have reached practical upper limits on data set size which constrains our ability to adequately represent and accurately analyse enzymes that have been evolving for billions of years. We have therefore developed the tool Graphical Representation of Ancestral Sequence Predictions (GRASP) to take advantage of the rapidly expanding databases of known protein sequences and the information from biological diversity that can be mined from large protein families.

However, the challenge does not end with scaling-up processing capacity. Remote homologs that form part of large protein families are likely to have resulted from numerous evolutionary events, the order of which is often difficult to infer, and confounds current phylogenetic analysis techniques; insertion and deletion (indel) events introduce dependencies between positions in sequence that make their history difficult to access at greater scale [12, 13]. In practice, their presence has been shown to decrease alignment accuracy unless appropriately handled [14, 15]. Lee et al. [16] demonstrated how a partial order graph (POG) can be used to represent and support the alignment of widely different sequences. The risk of aligning sequence fragments with different evolutionary origins motivated us to use the POG data structure to enable the separation of distinct sources of sequence diversity at evolutionary branch points.

The premise of our study was that a substantial increase in the scope of ASR will provide (a) a rich resource for evolutionary studies, and (b) valuable guidance for protein engineering,

given the demonstrated usefulness of ancestral enzymes as robust templates for directed evolution [8]. Accordingly, our method was designed with a view to identifying substitutions, insertions, and deletions that may be combined to form sequence configurations inspired by, but not necessarily present in, either extant or inferred ancestral sequences. Recent advances in protein engineering have shown this to be an effective way of surveying sequence space and here we applied similar approaches explicitly to ancestors [17]. In particular, we sought to evaluate if inferred sequences and positions of indels can be used as building blocks to engineer biologically active ancestors. This, in turn, demands that we first identify a practical approach for their inference, in parallel with substitutions.

In brief, this paper demonstrates the capacity of an approach based on POGs and maximum likelihood inference to perform ASR of large protein families and assist in the design of novel biocatalysts. Specifically, we explored the impact of quantity, diversity, and taxonomic context of input sequences on predicted sequences as well as resurrected structures and functions. Critically, we evaluated the prospect of re-purposing indel events inferred from the evolutionary record to compose and resurrect "hybrid" ancestors.

Experimentally, we resurrected ancestral proteins from three families (exemplifying various degrees of sequence number, functional diversity, and sequence similarity) and evaluated these in terms of their structure and function. First, the glucose-methanol-choline (GMC) oxidoreductases represent a super-family of enzymes with varying biological functions and industrial applications; we focused on the enzymes glucose dehydrogenase (GDH, EC 1.1.5.9) and glucose oxidase (GOx, EC 1.1.2.4) [18]. Second, members of cytochrome P450 subfamily 2 (CYP2) play a key role in drug and xenobiotic metabolism in metazoans [19]. Here we concentrated on the CYP2U subfamily and two closely-related subfamilies, CYP2R, and CYP2D. Thirdly, the IlvD/ED dehydratase family includes dihydroxy-acid dehydratase (EC 4.2.1.9) and several sugar acid dehydratases all containing iron-sulfur-clusters across a broad taxonomic scope [20]. We refer to this family as DHAD. It is present in bacteria, archaea, fungi, algae, and in some plants. We also evaluated the large-scale inference of, but did not resurrect, the ketol-acid reductoisomerase (KARI) family, which includes enzymes in the branched-chain amino acid biosynthetic pathway (similar to DHAD) present in bacteria, fungi, and plants. We focused on KARI class I for a large-scale inference, and class II for a comparison between existing tools, having previously successfully resurrected ancestors of class II enzymes [8].

## Results

### GRASP infers partially ordered ancestor graphs, representing substitutions, insertions, and deletions

GRASP implements maximum likelihood for inference of ancestor states within trees, which in turn are graphical models [21]. It uses efficient algorithms developed for Bayesian networks [22] to process volumes of data that are unprecedented in the field of ASR.

Unlike other reconstruction methods, GRASP uses POGs to represent sequence content, including insertions and deletions, to allow for (1) inference and visualisation of indel events across ancestors, and (2) representation of candidate ancestors as alternative paths through an ancestor POG, as described in Methods and illustrated in Fig 1.

GRASP infers ancestor POGs from an input POG that represents a set of aligned homologous sequences and an input phylogenetic tree describing their evolutionary relationships. It does this in three stages that are designed to deconvolute sources of sequence variation.

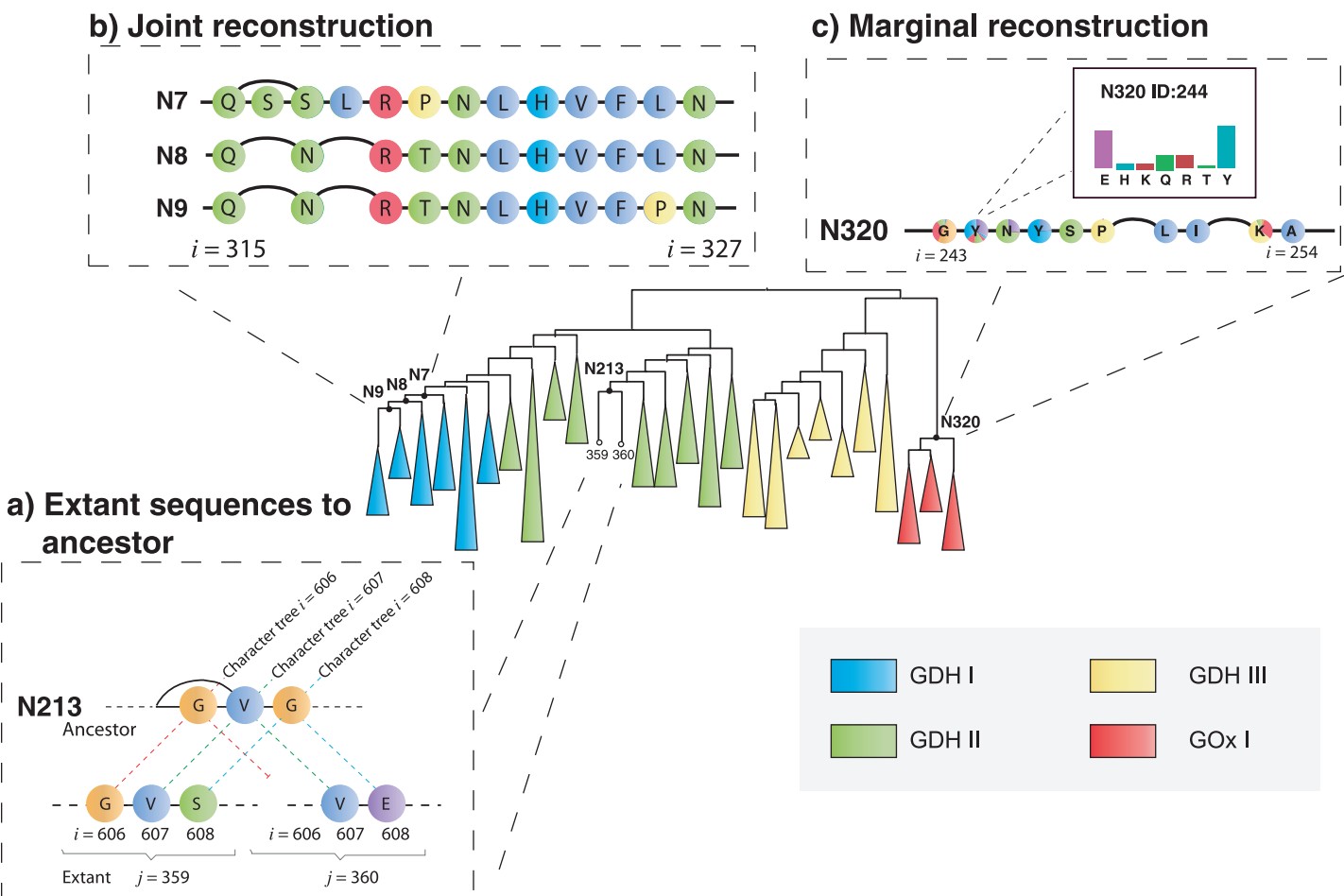

**Fig 1. Phylogenetic tree showing a reconstruction of fungal GDH and GOx sequences decorated with illustrations of key concepts used within GRASP. A**, Two extant POGs ($j$ indicates extant sequence number) mapped to an ancestral POG. Each extant POG has a single path through strictly ordered sequence positions ($i$ indicates position). Ancestral states are influenced by all sequences, which explains why $i = 608$ is inferred as glycine, despite glycine not appearing in either sequence $j = 359$ or 360. **B**, Three ancestor POGs showing most probable assignments from a joint reconstruction at positions $i \in \{315, \ldots, 327\}$ for nodes N7, N8, and N9. GRASP supports the simultaneous viewing of multiple ancestors from a joint reconstruction, enabling a direct comparison at different time points. **C**, A single ancestor POG showing inferred marginal distributions at positions $i \in \{243, \ldots, 254\}$ for node N320. For marginal reconstructions, nodes are coloured according to their posterior probabilities and can be queried to view histograms of these underlying distributions, as is done for position $i = 244$. The marginal reconstruction from (c) was used to reconstruct the inferred ancestor (N320) as well as an alternative ancestor in which a single amino acid (N320_Y244E) was altered based on posterior probabilities from the marginal distribution that resulted in increased thermal stability (S1 Table).

1. A history of indel events is inferred by either maximum likelihood or maximum parsimony and mapped onto the tree to determine which positions contain actual sequence content (Fig 1A).

2. For each ancestral position, the most probable character (to explain those observed at the leaves) is assigned to each phylogenetic branch point when performing a *joint reconstruction* (Fig 1B). Alternatively, for each position at a nominated branch point, the probability distribution over all possible characters is inferred when performing a *marginal reconstruction* (Fig 1C).

3. Edges are drawn to represent *all* inferred combinations of indels to form an ancestor POG with nodes that can form a valid sequence with inferred content; a preferred path through the POG is then inferred, nominating a single, best supported sequence.

For GDH and GOx, we used GRASP to identify potential substitution variants through analysis of inferred distributions via marginal reconstruction. Marginal reconstruction is frequently performed to account for uncertainties in reconstructed sequences, suggest alternative or variant ancestors, and explore properties such as thermal stability or substrate preference in inferred variants (Fig 1C, S1 Table).

## GRASP implements novel options for inferring indel histories

Insertions and deletions at branch points introduce a major source of variation in sequence. Not all reconstruction tools model indels, but several consider each sequence position independently and infer "gaps" in ancestors through parsimony (PAML/LAZARUS [23]) or by probability based on the presence of gaps in descendants (GASP [24], FireProt[ASR] [25]). By referencing extant and ancestor sequences in a multiple sequence alignment, extended gaps are accounted for by simple indel coding (SIC) that regards deletions as equivalent if they start and end at the same positions [26]. This in turn means that gaps have a binary state; they are either present or absent in a sequence. When gaps are considered relative to another, a third option "inapplicable" applies to a gap that is "contained" by another, thereby allowing preferences to be followed when multiple gap states are possible. It is therefore possible to encode a set of deletions by SIC and infer the optimal presence of each deletion independently at evolutionary branch points by parsimony or maximum likelihood (FastML [27]).

GRASP implements three methods for encoding patterns of insertions and deletions from the set of input sequences: position-specific (PS) encoding, SIC, and a novel method that considers indels as edges in a POG, which we refer to as bi-directional edge (BE) encoding. Similar to SIC, BE encodes indels by their start and end positions in a multiple sequence alignment. However, BE does not treat indels with either shared start *or* shared end positions independently when viewed from the alignment. Indeed, BE allows the inference to differentiate between deletions that start at the same position but end at different positions (forward direction) as well as to differentiate deletions that end at the same position but start at different positions (backward direction). From the point of the resulting (unaligned) ancestor sequence, GRASP basically determines (for each site) the best neighbours both to its right and its left. Every indel event can then be designated as absent from the ancestor or present, and deemed to be optimal in one, both, or neither direction.

GRASP implements both parsimony and maximum likelihood to infer indel histories based on any of the three encodings, giving six indel inference options (PS-P, PS-ML, SIC-P, SIC-ML, BE-P, and BE-ML). For all options, GRASP represents ancestors as POGs, which can be used to identify the most supported path, which in turn is a contiguous sequence.

## Bi-directional edge methods return fewer unique indel events

It is not obvious to what extent predicted indel variants differ at ancestral branch points in a given evolutionary tree, based on the choice of encoding and inference method. Moreover, given the unavailability of historical data, we are limited to evaluating the accuracy of predicted indel events on simulated data. We used INDELible [28] to generate indel histories at four indel rates (rate of indel events relative to substitution events) and for four taxon sizes (number of extant sequences within each data set), which resulted in phylogenetic trees and sequences with identified gaps at all branch points and leaves.

These parameters gave alignments with an overall percentage of gaps that were either comparable to or contained more gaps than the non-simulated data within this study (S1 and S2 Figs).

In each run, these gapped sequences form an alignment where all positions are guaranteed to be homologous regardless of the number of indel events (S3 Fig). The combination of tree and alignment from INDELible allows indel events (defined by start and end position) to be tracked and retrieved from branch points to measure the accuracy of the reconstruction at different rates and taxon sizes. We defined accuracy as the percentage of correctly recovered indel events (by position and branch point). The simulation of each setting was repeated to render different data sets to determine the variance explained by random factors.

All six indel inference options recovered the correct length of ancestor sequence at the root, for each taxon size and indel rate, across every replicate. Accuracy was consistent and variance decreased as taxon size increased for a given indel rate, and accuracy decreased as the indel rate increased (Fig 2A). The number of indels that were consistently identified or consistently missed by all methods stayed constant as taxon size increased for a given indel rate. As the indel rate increased, the number of consistently identified and consistently missed indels decreased and increased, respectively (Fig 2B).

As all methods performed comparatively and to simulate more challenging and realistic scenarios, we realigned the sequences generated by INDELible with MAFFT; we then used this realigned data with the original trees to generate additional sets of ancestors. In this scenario, we could no longer directly measure accuracy because the re-aligned sequences probably implied indels with different start and end positions. We therefore evaluated the effect of indel rate and taxon size on the sequence length at root and qualitative differences in the sets of indels identified by each indel inference method. This in turn would enable us to understand which indel inference method offers the broadest support in the pursuit of indel variants for engineering applications.

Under the realigned conditions at the highest indel rate, SIC returned root ancestors with longer sequence length regardless of the inference method or taxon size (S4(A) Fig). There was substantial overlap in indels predicted by each method. That said, BE methods identified fewer *unique* indels than PS or SIC methods (see S4(B) Fig for an example replicate from the largest taxa size with the highest indel rate). The number of unique events predicted by all methods increased with both indel rate and taxon size, but BE methods tended to identify fewer. More-over, BE was more often in agreement with at least *one* other method, compared to either SIC or PS. PS methods and SIC-P returned higher numbers of unique one-residue indels and SIC methods returned higher numbers of unique indels of length greater than three (Fig 2C and S5 Fig). Unlike BE, SIC focuses on the representation of gaps rather than the edges that make up a sequence. SIC assumes that the starting point is a linear sequence, on which gaps are super-imposed. SIC does not inherently resolve conflicts between multiple, partially overlapping indels, which are independently predicted at each branch point. POGs do not explicitly repre-sent gaps, allowing predicted, conflicted indels to be separated. The conversion to a linear sequence involves choices that benefit from considering the evolutionary context and can be ascertained across multiple branch points in the phylogenetic tree.

FastML implements SIC and offers both maximum parsimony and maximum likelihood inference, the latter with a gene loss/gain model.

We used parsimony inference to ensure that the implementations were comparable, noting that FastML presents a single state for each indel (independent of all other indels) as evident by the ancestor sequence. The flexibility of ancestor POGs enables GRASP to present both presence and absence of indel when both are deemed optimal (still independent of all other indels).

While FastML infers sequences and GRASP infers POGs, we tested at each branch point if the POG contained a valid path for the sequence predicted by FastML. For INDELible trees with 50 extants and 50 positions, even at the highest indel rates, SIC-predicted POGs

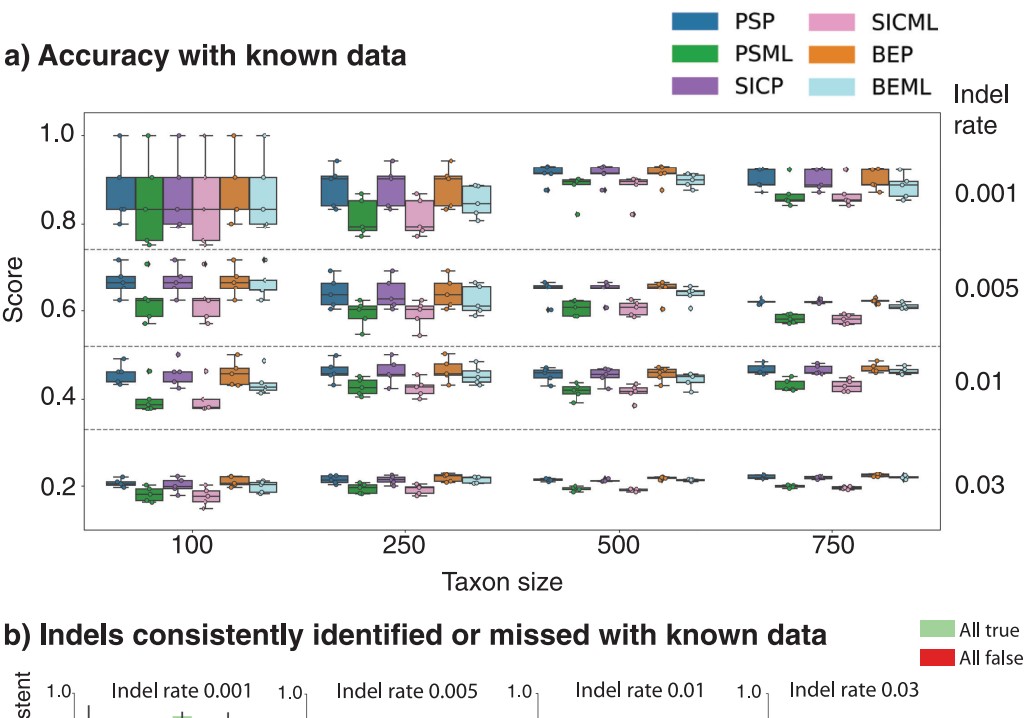

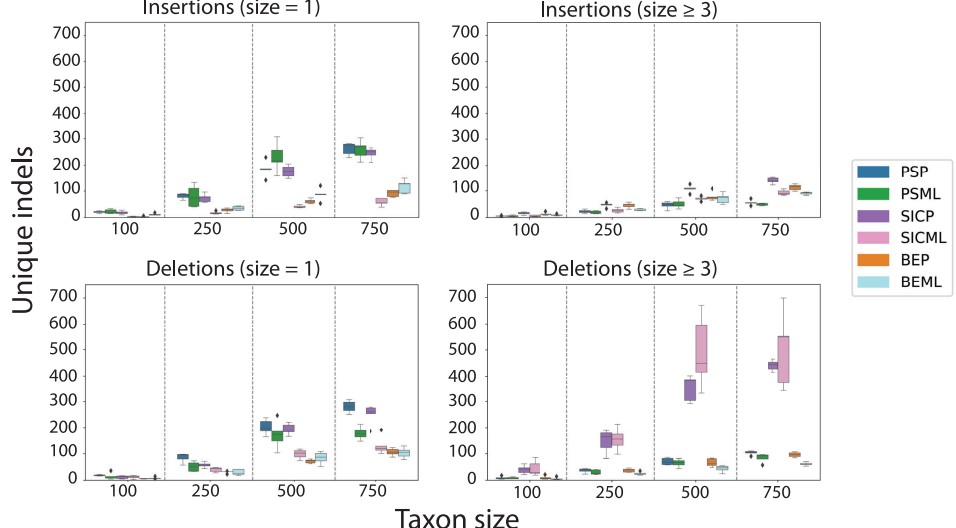

**Fig 2. Results of indel evaluation of each of GRASP's indel methods across 1) simulated data where the correct alignment and correct tree are supplied and 2) simulated data where a realigned alignment and correct tree are supplied across four indel rates (0.001, 0.005, 0.01, 0.03) and four taxon sizes (100, 250, 500, 750). A**, Number of correct indels identified by each method for each taxon size for each indel rate ($N = 5$). **B**, Number of indels uniformly identified by all methods or uniformly missed by all methods for each taxon size for each indel rate ($N = 5$). **C**, Number of indels uniquely identified by each indel method at four taxon sizes at indel rate 0.03, organised by indel type and size ($N = 5$).

contained the corresponding FastML prediction. At 100 extants and longer sequences, a small number of deviations occurred, which we expect are due to restrictions imposed by mapping inferred encodings to a valid POG.

Unexpectedly, the indel tests with known alignments where ancestral indel events could be tracked precisely showed little difference between encoding and inference method. With non-curated alignments, there is a greater difference between methods. To next evaluate if insertions and deletions can be used as building blocks to engineer biologically active ancestors, we opted to use the BE encoding, which was better at capturing what the others also captured while containing less outlier indels; we opted for parsimony inference, which seemed to perform comparably to maximum likelihood but was generally quicker.

## Indel variation can be used to create hybrid ancestors

The insertion or deletion of sequence blocks is well established as a means by which natural proteins evolve new functions. However, previous attempts to use insertions and deletions in protein engineering have been confounded by the fact that indels frequently compromise protein integrity. The location of the insertion or deletion determines its tolerability, but suitable locations are not necessarily evident from the structure and even less so from the sequence alone.

We hypothesised that indel events suggest plausible blocks of sequence content that could be included or excluded in identified ancestors as a novel approach to creating ancestral variants, orthogonal to substitution. GRASP utilises the history of indel events to predict modular blocks of content capable of being removed from ancestors in which they occur or inserted into ancestors that never contained these modules. In doing this, GRASP fundamentally extends the nature and practical application of modulating variation within ancestors and is capable of identifying modular insertions that are well tolerated and can affect properties such as thermal stability and substrate specificity. The ability to manipulate both of these properties is desirable for protein engineering.

We used GRASP to identify two distinct lineage-specific insertions within the CYP2U/CYP2R/CYP2D data set (LSEE, LLSPP), occurring at nodes N5 and N51 (Fig 3). We synthesised the inferred ancestors N5 and N51, as well as a more ancient ancestor, N2, that did not contain either insertion. We removed the insertion LSEE from N5 at sequence position 153 (N5_153dLSEE) and removed the insertion LLSPP from N51 at sequence position 27 (N51_27dLLSPP). We then separately inserted them into N2 at the equivalent sequence positions 152 (N2_152iLSEE) and 27 (N2_27iLLSPP). Similarly, we tested two variants of the N1 CYP2U ancestor. One form contained a CYP2U-specific insertion of 19 amino acids (N1), and in the other this insertion was removed to resemble the CYP2R and CYP2D sequences (N1_18dIPP. . .RR).

All ancestral proteins inferred via this process were heterologously expressed in *Escherichia coli* and characterised. They were shown to express at similar levels and form intact haem-thiolate linkages as indicated by the characteristic spectral peak at 450 nm in the Fe(II).CO vs. Fe(II) difference spectrum (S6 Fig), indicative of a properly folded cytochrome P450 enzyme. All were catalytically active towards at least one substrate, when tested with three different P450-Glo luminogenic probe substrates, luciferins MultiCYP, ME-EGE, and CEE. Therefore, the presence or absence of these lineage-specific insertions was not essential for the protein folding, cofactor binding, or interaction with the cytochrome P450 reductase. However, it was observed that the lineage-specific insertions did alter the substrate selectivity of the otherwise identical ancestors.

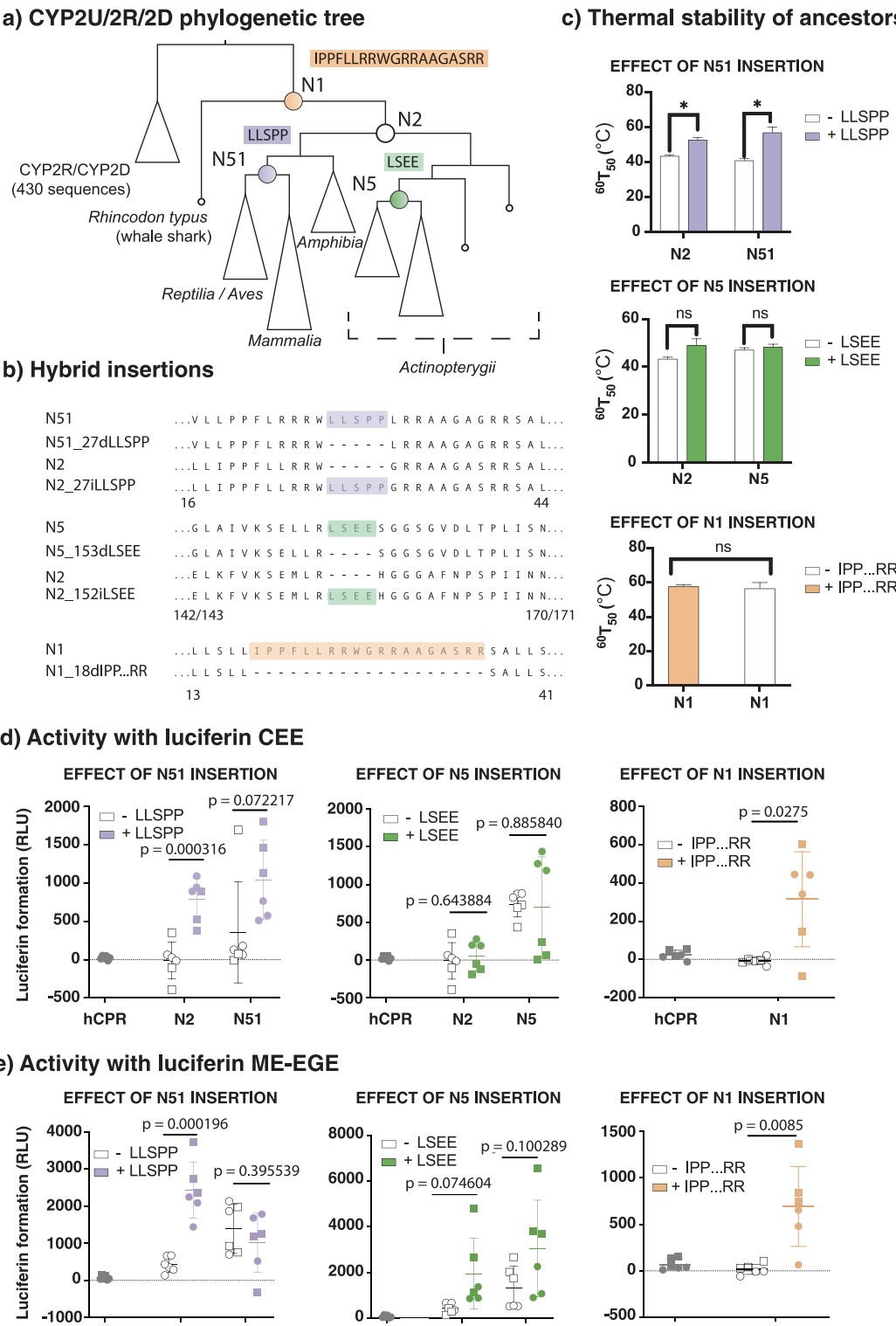

**Fig 3. A, Phylogenetic tree showing positions of CYP2U/CYP2R/CYP2D ancestors chosen for synthesis and evaluation.** Coloured boxes indicate the content that was removed from correspondingly coloured ancestral nodes (N51, N5, and N1) and, in the case of the N51 and N5 insertions, preemptively inserted into N2. **B**, Amino acid sequences surrounding and including the insertion or deletion of the content at each ancestor. Numbers under sequences indicate the position numbers of the start and end columns represented in the alignment. **C**, Thermal stability assays for each ancestor with and without inserted content. Data are means +/- SEM, $N = 2$, ns indicates a not significant result. **D, E**, Activity assays

for the substrates luciferin CEE and luciferin ME-EGE for each ancestor with and without inserted content. Membranes from cells expressing only human CPR are included as a negative control. Different symbols indicate different experimental repeats, performed in triplicate, lines indicate mean and standard deviation, and p-values were determined by a two-tailed Student's *t*-test. No data points were excluded.

Both N5 and N51 were active towards both luciferins CEE and ME-EGE, while N2 was only active towards luciferin ME-EGE. Loss of the insertion LLSPP from N51 reduced its activity towards luciferin CEE, and the corresponding gain of the LLSPP insertion in the N2 ancestor increased its activity towards luciferin CEE. Neither loss of the insertion LSEE from N5 nor gain of the insertion LSEE in N2 had an effect on luciferin CEE dealkylase activity. The presence of the LSEE insertion in the N2 and N5 ancestors increased both ancestors' activity towards luciferin ME-EGE. Inclusion of the LLSPP insertion did not have a consistent effect on activity towards luciferin ME-EGE, whereas inclusion increased activity towards ME-EGE in N2, but not in N51. The N1 ancestor was only active towards luciferin MultiCYP, but N1_18dIPP...RR was slightly active towards all three pro-luciferin substrates, suggesting this insertion may also alter the selectivity of the ancestor.

The LLSPP insertion also modulated the thermal stability of the ancestors; the insertion produced a small but statistically significant increase in the thermal stability in both the N2 and N51 ancestors, compared to their variants lacking this insertion (Fig 3C). This effect was not seen for the LSEE insertion (Fig 3C), suggesting that these effects are protein and sequence specific.

The major alpha helices in the cytochrome P450 fold are named sequentially from A-L, following the protein backbone [29]. Both insertions discussed here occur within loops in either the AA loop (LLSPP insertion) or the D-E loop (LSEE insertion) of CYP2U. The AA loop occurs near the N-terminal region of the protein and is thought to be involved with protein-membrane interactions. The D-E loop is a surface loop and is not located near the structurally important regions, such as the haem binding pocket, the CPR-interaction interface and the putative substrate entry channel [30, 31]. Both the AA' loop and the D-E loop are both surface loops positioned on the distal side of the haem cofactor but while the AA' loop is located near the transmembrane helix and is expected to be partially embedded in the membrane, the D-E loop is structurally distant from the AA' loop [30, 31] (S7(A) Fig).

The effects of these ancestral insertions were investigated using models generated using AlphaFold2 run via ColabFold. [32, 33] (S7(B) Fig). However, due to poor confidence in the AA' loop region prediction (S7(C)–S7(E) Fig), the structural models did not yield any additional insights into a role of loop truncation in CYP2U thermal stability. Similarly, only minor changes to the D-E loop were observed for the ancestors bearing the LSEE insertion (S7(F) Fig).

Finally, we considered whether the indels in CYP2U ancestors resulted from changes to the intron/exon boundaries in the CYP2U1 gene but were unable to find strong evidence either for or against this hypothesis.

## Using more sequences reduces variance in inferred ancestral sequences

GRASP is able to process large numbers of sequences (i.e., greater than 10,000), which is necessary to capture the true diversity of the sequence space. Intuitively, more data equates to better coverage (and resolution) of the biological sequence space. If this saturation of data is based on true homology, the level of variance is reduced in inferred ancestral states relative to reconstructions from less data. This intuition has been shown to be supported in experiments on

smaller data sets [34]. Here, we generalise this assertion for a much larger data set that contains the added complexity of obscured homology, due to a greater presence of indel events.

Indel handling is critical for ASR, yet routinely problematic, and the accurate management of indel events is essential to decide on the extent of sequence content to include for any particular ancestor [14]. As data set sizes grow, the number of columns in a sequence alignment, or positions in a POG, increases substantially and the indel histories may appear more complicated. Therefore, increasing the number of sequences does not necessarily lead to inferences of increasingly invariant ancestral sequences.

Evidence that ancestors are constrained to a specific invariant form as the number of sequences increases must be combined with evidence that this form is viable and not a sequence that has been converged upon arbitrarily. To test the effect of increasing data set size on ancestral inference, we assembled sequence data sets for the DHAD and CYP2U protein families via increments of sequence data (see Methods) and compared the ancestral inferences for each data set size (Fig 4A–4D), ranging from 1,612 to 9,112 sequences for DHAD and between 165 and 595 sequences for CYP2U. The DHAD data sets were increased by adding sequences from across the DHAD taxonomic space, while the CYP2U data sets were increased by adding sequences from sister groups CYP2R and CYP2D while retaining the same number of CYP2U sequences at each point. For the DHAD data set we also performed a sparse reconstruction of 585 sequences, containing primarily reviewed Swiss-Prot sequences.

With GRASP, we observed that as data set size increased, the predicted ancestor sequences approached invariant forms in terms of amino acid sequence at equivalent phylogenetic nodes between different tree sizes. To further illustrate these trends, we inferred KARI ancestors in regular increments ranging from 1,176 sequences to 11,756 sequences. These ancestors also converged towards invariant forms with the addition of more sequences (Fig 4E). While the number of positions in the input sequence alignment generally increased with coverage, the length of the ancestor sequences was not correlated with the number of input sequences (S8 Fig).

GRASP was able to complete the reconstruction of the largest data sets in this study within 7 hours for DHAD (9,112 sequences, 1,381 positions in alignment) and within 6 hours for KARI (11,756 sequences, 667 positions) (S9 Fig).

Ancestral proteins inferred from the smallest and largest data sets for both DHAD and CYP2U were all active towards expected substrates, despite differences in ancestral sequence identity between the two extremes of data set size (DHAD 75%, CYP2U 80% sequence identity). All DHAD ancestors displayed enzymatic activity to D-gluconate. We observed that three DHAD ancestral proteins from the smallest data set had thermal shift profiles comparable to those of the three ancestors that were located in equivalent tree positions in the largest data set (S10 Fig). For two of the three DHAD reference ancestors, the melting points in the proteins from the larger reconstruction were increased by approximately 5°C relative to their counterparts from the smaller data set (S10 Fig). Likewise, the inclusion of the sister clades for the CYP2U reconstruction increased the thermal stability of the CYP2U whole-subfamily ancestors inferred from each set (165, 359, and 595 sequences) and these ancestors were all shown to be active towards the substrate, luciferin MultiCYP (S11 Fig).

## On smaller data sets, GRASP's predictions are consistent with the predictions of existing methods

We compared GRASP against two alternative ASR tools, selected due to their dominant use in the literature: FastML [35] and the aaml program from the Phylogenetic Analysis by Maximum Likelihood (PAML) package [36]. We were able to produce ancestral proteins from

## a) DHAD phylogenetic trees of 1612 vs. 9112 sequences

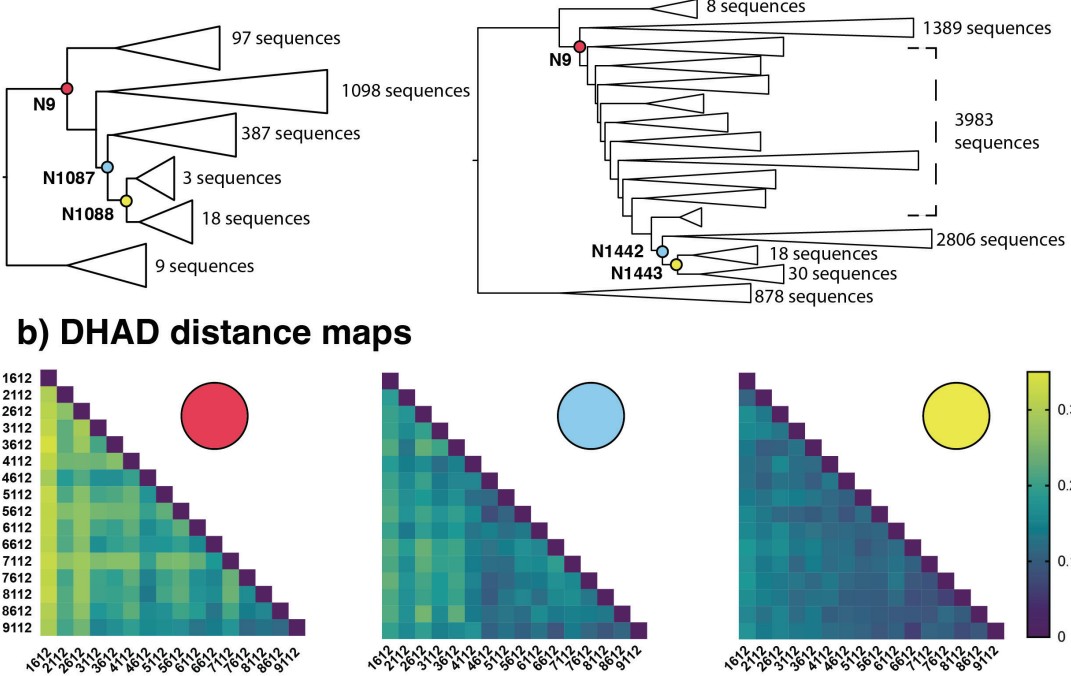

## b) DHAD distance maps

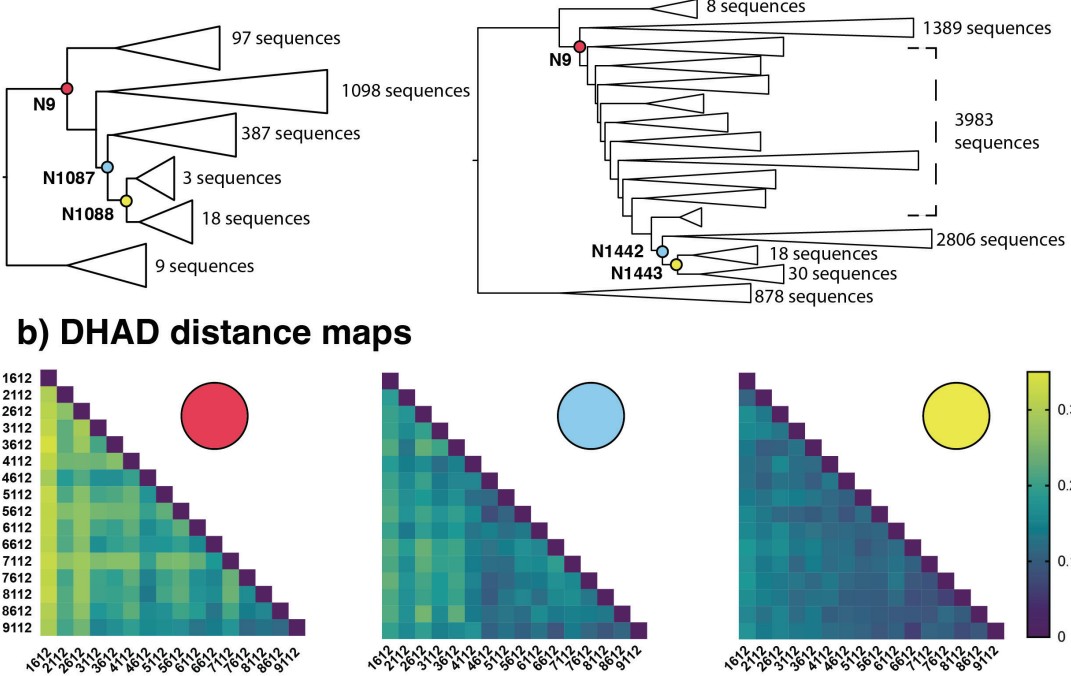

## c) CYP2 phylogenetic trees of CYP2U (165 sequences) vs. CYP2U/CYP2R/CYP2D (595 sequences)

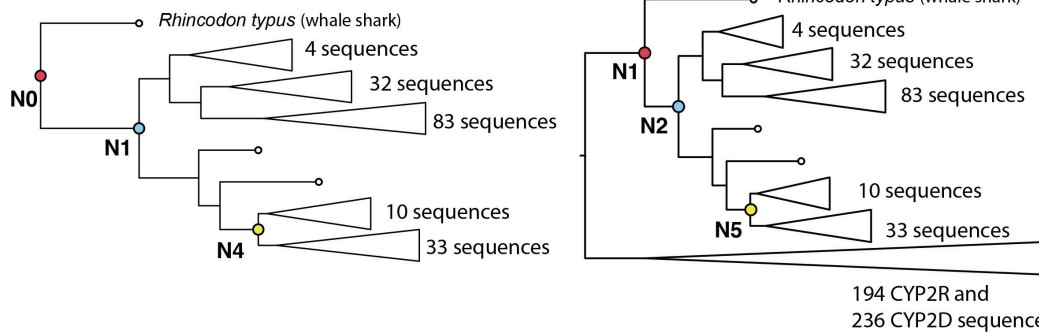

## d) CYP2 distance maps

## e) KARI distance map

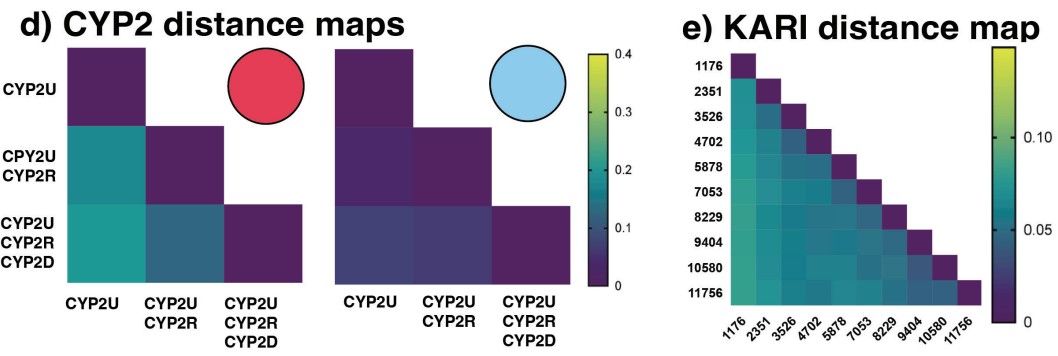

**Fig 4. A,** Phylogenetic trees of the smallest and largest DHAD data sets after producing 14 randomly sampled data sets in 500 sequence increments, added to our base data set of 1,612 sequences and reaching a maximum size of 9,112 sequences. **B,** Heat maps of the fractional distances between ancestor sequences generated from different DHAD data set sizes, representing the same (principal) three branch points. **C,** Phylogenetic trees of the smallest and largest data sets after increasing CYP2U sequences via addition of homologous subfamilies, starting with 165 CYP2U sequences then growing to 359 sequences and reaching a maximum of 595 sequences via addition of sequences from CYP2R and CYP2D, respectively. **D,** Heat maps of the

fractional distances between ancestor sequences resulting from different CYP2U data set sizes representing the same two branch points. Ancestors from the N4/N5 equivalent branch points across the three data set sizes had 98% identity, which cannot be discerned visually. **E**, Heat map of the average fractional distance of 50 randomly selected ancestors between the KARI I data sets, ranging from 1,176 to 11,756 sequences.

reconstructions produced by GRASP, FastML, and PAML on a CYP2U/CYP2R data set comprising 359 sequences. To compare between indel inference methods again, we noted that GRASP (BE) and FastML (SIC) returned the same indel composition at their root ancestors and this pattern was overlaid onto the prediction from PAML, which was run without gap inference. The oldest CYP2U ancestors had ∼95% sequence identity and, regardless of the tool used, ancestral proteins expressed at similar levels in *E. coli*, displayed characteristic cytochrome P450 spectra and activities towards the luciferin MultiCYP substrate, and had similar thermal stabilities (S12 Fig).

To make statements about the accuracy of ancestral predictions is problematic as the historically correct and complete evolutionary record is unavailable. To sidestep this issue, we first applied each tool and configuration to generate multiple predictions of the *same* principal ancestor branch point based on successively smaller subsets of a given sequence family. Secondly, we performed two tests asking: (a) between tools, how similar is the prediction of one tool to those of the others; and (b) how similar is the prediction of one tool from the sub-sampled data to a better-sampled ancestor, predicted from the *complete* family? We reasoned that a better method would be one which tended to agree with the majority of others, and one that with *less* data tended to agree with a prediction based on *more* data (assuming that more data help to improve a prediction).

A large alignment of 1,682 sequences (KARI class II, adapted from Gumulya et al. [8]) and the corresponding phylogenetic tree were divided into sub-groups and used to assess the effect of tool, data set size, and reconstruction parameters on ancestral inference (see Methods for details). We sought to corroborate any trends using a second independent data set (CYP2 with 975 sequences).

Test (a) measured similarity between ancestors inferred using a given set of tool parameters and group size (Fig 5A and S13A Fig); specifically, we observed fractional distances $D/L$ (where $D$ is the number of substitutions, of $L$ non-gapped, homologous positions) between sequences predicted for each condition tested. Test (b) measured similarity in terms of fractional distances between ancestor predictions of an individual tool (with a given set of parameters and group sample size) and a better-sampled ancestor using all available data (Fig 5B and S13B Fig). The better-sampled ancestor for the comparison in (b) was predicted by GRASP, because a data set of this size could not be completed by FastML or PAML. A series of statistical tests were performed. First, ANOVA was used to evaluate whether the choice of tool, data set size, and rate parameter setting were factors in determining how similar a predicted ancestor sequence (grouped by tool/size/rate setting) was to those predicted by alternative tools with the same setting (test a) and to those generated from the complete data set (test b). A *t*-test was then used to identify the pairs of labels (on groups) that best explained the observed differences (S14 Fig and S15 Fig).

While GRASP can perform character inference via variable evolutionary rates when estimated externally, we opted to use the default uniform rate. Both FastML and PAML estimate position-specific rates internally, so we set out to understand the impact of rate setting. When comparing predictions between tools (test a) or between tool and better-sampled ancestor (test b), both the choice of tool and data set size separately and consistently explained the observed differences in distances; however the rate setting did not.

## a) Distance between tools' ancestors

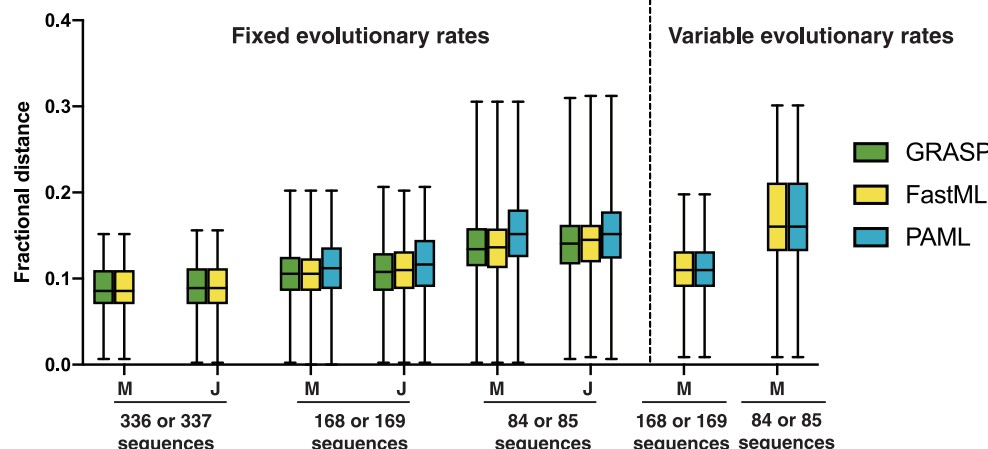

## b) Distance between a tool's ancestor and better-sampled ancestor

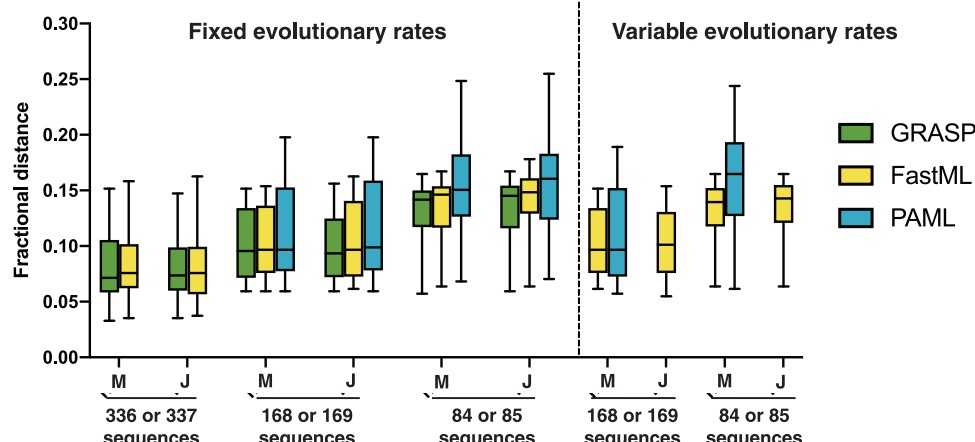

## c) Run times for GRASP and FastML

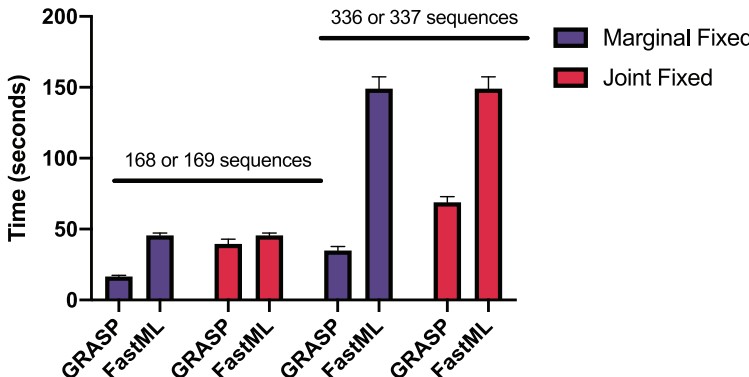

**Fig 5. Tool comparison on KARI data. A**, Average fractional distance between tools, calculated as pairwise fractional distances for each ancestral prediction for a given tool against all other ancestral predictions of other tools using 5 groups of 336 or 337 sequences, 10 groups of 168 or 169 sequences, or 20 groups of 84 or 85 sequences. Parameter choices are joint (J) vs. marginal (M) reconstruction and fixed vs. variable evolutionary rates (FastML and PAML only). **B**, Average fractional distance between a better-sampled ancestor inferred by GRASP using 1,682 sequences and each tool / parameter combination at 5, 10, and 20 groups. **C**, Run times of GRASP and FastML at 5 and 10 groups; PAML was omitted due to long run times. Run times for all tools at 10 and 20 groups are shown in S16 Fig.

The choice of tool mattered for both types of comparisons across the two data sets. GRASP and FastML predictions were generally more similar in terms of mean pairwise fractional distance, both relative to the better-sampled ancestor and relative to the ancestors predicted by PAML. Indeed, greater sequence numbers generally reduced distances between tools and the distances between a tool's predictions and the ancestor based on the complete data set.

We also calculated the time taken for all tools to complete the reconstructions with a run time cut-off of 48 hours (S16 Fig) and highlighted the time taken for GRASP and FastML to complete the two larger data set sizes (Fig 5C).

## Discussion

Increasing the scale at which ASR can be performed means that greater sequence and functional diversity can be explored and more complicated phylogenetic relationships can be evaluated [11]. Incorporating data from sister clades and remote homologs can allow ancestors to be inferred with greater robustness at greater evolutionary distances. We substantiated these points computationally and experimentally by resurrecting twenty ancestral variants across three enzyme families, all of which were shown to be catalytically active.

At greater scale, ASR will increasingly require judgements to be made as to whether ancestors contain an insertion or deletion present in only some of the multiple descendent branches of a phylogenetic tree. The presence of these events may pinpoint loop remodelling events [37] or other conformational diversity in the family. The evolution of conformational diversity may promote entirely new functions and provide a source of variation that protein engineers can exploit [9, 10].

At the core of our approach is the POG data structure, originally proposed to facilitate multiple sequence alignment [38, 39]. We developed inference methods to directly use the POG data structure to pinpoint likely phylogenetic positions for indel events from homologs placed in an alignment. It effectively delineates sequence content at all internal nodes of a given phylogenetic tree, and collectively enables the evolutionary relationships between all branch points to be traced. As a consequence, evolutionary events that are isolated to specific clades, and alignment ambiguities that are difficult to resolve at a single branch point, can be disentangled across time. In contrast to current approaches based on gapped sequence representations, POGs enable the identification of all supported indel histories across a reconstructed family. In GRASP, edges that are partially supported but are not chosen to form the preferred ancestral sequence still appear as alternative paths through the POG; multiple branch points can be inspected and contrasted at once. GRASP thus provides a framework to delete, reintroduce, or preemptively include indel variation that supports both exploring sequence space and creating new function.

Hybrid ancestors represent a novel class of variant that can be identified readily and resurrected through the partitioning of indel events onto individual edges within a POG. The identification of alternative indels, each compatible with a given ancestor (or nearby ancestors) can be used to test alternative hypotheses about the progression of evolutionary events where there is uncertainty in the inferred evolutionary history. In addition, combinations of compatible indels can be sampled in order to engineer novel sequences by including or removing blocks of sequence content. Strikingly, given the substantial impact that indel events are likely to have on any protein sequence, coupled with the divergence between chosen ancestral sequences, all hybrid ancestors we reconstructed were functional.

With CYP2U as an example, we showed that hybrid ancestors folded to form holoenzymes that are catalytically active when tested *in vitro* and are capable of interacting with the native human reductase. Additionally, with CYP2U, we showed that inclusion of these modular

blocks allowed for increased thermal stability and altered substrate preference. We stress that we are not attributing the increased stability or interaction with a specific substrate solely to the identified insertion. Rather, GRASP identified blocks of content that are likely to be tolerated at positions at which such changes have occurred during natural evolution, despite any effect they have on folding and function of these ancestral proteins. Due to the complex nature of protein folding and epistasis, these blocks will not always behave in predictable ways and effects will depend on the ancestral sequence and sequence context into which they are being inserted.

The identification of modular insertions altered the substrate selectivity, not unlike how substitution variants identified by marginal reconstruction have in previous studies [8]. This study provides a proof-of-concept that indel histories can suggest a form of variation that protein engineers can use that is orthogonal to varying specific amino acids. We foresee this as being of practical use for (1) altering function through the addition and removal of discrete, evolutionarily-defined building blocks to engineer variants with altered catalytic and physical properties (e.g. thermal stability) and (2) exploring alternative ancestors where there is ambiguity in the true phylogenetic position of an indel.

There are multiple options available for predicting indel events, and on simulated data they perform equally well. Bi-directional edge methods, which consider indels in the context of other possible choices, allow more opportunities to decide between alternative pathways and provide better ways to represent ambiguity when equally parsimonious or equally likely scenarios exist. This may prove beneficial to the protein engineer, who may have access to secondary data to guide their selection for applied purposes, including knowledge of structure or residue interaction.

ASR has been used extensively in recent years [4, 9, 11] and it is important to understand the relative performance of different tools, and to recognise proven principles that underpin successful methods. We note that FastML is in broad agreement with GRASP, with predictions made by FastML showing closer evolutionary distances to predictions made by GRASP than to those generated by PAML. We also demonstrated that incorporating more sequence data resulted in smaller fractional distances between inferred ancestors, regardless of the tool used.

Based on the analyses of KARI and DHAD, we demonstrated that despite an increase in diversity, ancestral sequences converge towards invariant forms when using data sets with almost 10,000 sequences. Ancestral sequences generated from smaller data sets exhibit greater variation in ancestral sequence identity relative to the ancestral sequences from larger data sets. This supports the notion that greater representation of a family provides a constraint for the ancestor, i.e. that robust reconstructions are best achieved when available sequence data is exploited to the fullest extent.

## Methods

### GRASP implementation

The three main stages of GRASP are (1) to construct an indel history for every position in the alignment, (2) to infer character states for each ancestor, for all positions relevant to it (subject to step 1), and (3) to form a POG for each ancestor by linking all inferred character states (subject to steps 1 and 2) and optionally suggesting one path to generate a single sequence.

GRASP infers ancestor character states from a set of $M$ input sequences $\mathbf{S} = \{S_j : j \in \mathbb{J}\}$ where $\mathbb{J} = \{1, 2, ..., M\}$; $\mathbf{S}$ has $N$ aligned positions, indexed with $i \in \mathbb{I}$, where $\mathbb{I} = \{1, 2, ..., N\}$. In classical sequence alignments, positions without sequence content are padded, often shown as '–'; we use $\mathbb{I}^{(j)} \subseteq \mathbb{I}$ to index positions in sequence $j$ with actual sequence content, $S_{ji} = x$

where $x \in \mathcal{A}$ when $i \in \mathbb{I}^{(j)}$ and $\mathcal{A}$ is the set of amino acids. Later, it will be convenient to refer to $\mathbb{J}^{(i)}$, which indexes all sequences *with content* at position $i$.

Inference is based on a given phylogenetic tree $T$ with a nominated root, that has $M - 1$ branch points (if bifurcating, fewer when multifurcating) indexed by $\mathbb{K} = \{M + 1, M + 2, ..., 2M - 1\}$; we designate the index $k = M + 1$ for the root of the tree, by convention also labelled "N0" (followed by "N1", "N2", ... enumerated per depth-first in many tools including GRASP). The superset of extant and ancestor sequences (matched to POGs) is indexed by $\mathbb{Z} = \mathbb{J} \cup \mathbb{K}$. The topology of $T$ defines parent-child relationships, $\mathbb{Z}^{(k)} \subseteq \mathbb{Z}$ indexes the ancestral descendants of an ancestor $k$; conversely, we define a function $\kappa(k') = k''$ to indicate that $k''$ is the direct ancestor of $k'$, where $k' \in \mathbb{Z}^{(k'')}$.

Character states are inferred with an evolutionary model (in the form of an instantaneous rate matrix, indexed by $\mathcal{A}$) and maximum likelihood [40]. Inference is implemented via a Bayesian network that shares the topology of a position-specific tree, which is $T$ minus the ancestors without content at the position. The position-specific trees are constructed by an indel inference method (in turn informed by an indel encoding). While GRASP implements a number of different encodings, our focus below is on bi-directional edge encoding, which is new.

Below, we first define key data structures, then we distinguish between: (a) the handling of *where* homologous positions are placed relative to one another in the trace of ancestral sequences via POGs; and (b) the principles by which homologous positions in extant sequences are used to determine ancestral character states at branch points in the phylogenetic tree. The principles under (b) are unremarkable in themselves, but key benefits are achieved by using them in the ancestor POG from (a). For succinctness, we describe this procedure as it applies to a bifurcating tree; however, the same principles extend seamlessly to multifurcating trees.

## Representing sequence content as a partial order graph (POG)

A POG is a directed acyclic graph whose elements are ordered relative to other elements; a strict ordering is enforced *within* a subset of elements, but not always *between* subsets. When an order is imposed among elements, the relationship must be reflexive, anti-symmetric, and transitive [16]. A growing body of work in sequence alignment has demonstrated the flexibility that POGs offer for detecting and representing homologous sequence elements during alignment [38, 39, 41]. We take advantage of the flexibility of POGs when projecting homologous elements back in time; they represent deletions and insertions by edges that exclude or include alternative character subsets allowing for optional histories by offering multiple paths at ancestral branch points.

Formally, a POG is defined by a set of (up to) $N$ nodes that are indexed by $i \in \mathbb{I}$. The indices are determined by performing a topological sort on the "input POG" (see below); this gives at least one linear and complete ordering (out of several possible).

Nodes are connected by a set of directed edges, which is conveniently represented by a so-called adjacency matrix $\mathbf{E}$, where $\mathbf{E}(a, b)$ is set to 1 if there is an edge from $a$ to $b$, and otherwise 0; $a$ is topologically *before* $b$, meaning that a *uni-directional* graph can be represented by above-diagonal elements in $\mathbf{E}$.

We introduce an extended index-set $\mathbb{I}*$ for rows and columns in $\mathbf{E}$, with 0 and $N + 1$ to start and terminate the POG, so $a \in \mathbb{I}*$ and $b \in \mathbb{I}*$.

We define $next(\mathbf{E}, a) = \{b: \mathbf{E}(a, b) > 0\}$ and $prev(\mathbf{E}, b) = \{a: \mathbf{E}(a, b) > 0\}$ to refer to sets of nodes that occur after and before a node with a given index, respectively. $next(\mathbf{E}, 0)$ would thus give all possible start indices, and $prev(\mathbf{E}, N + 1)$ all terminating indices.

Moreover, we define *path*($\mathbf{E}$) to return all indices in $\mathbb{I}$ that can be accessed from 0 and $N + 1$, via recursive application of *next* and *prev*, respectively.

We distinguish between three types of POGs, the first two are determined directly from $\mathbf{S}$, and the third by inference. All POGs share the node index $\mathbb{I}$, which allows character states to be mapped across extant sequences and ancestors (illustrated in Fig 1).

- An "extant POG", is defined by a set of edges $\mathbf{E}^{(j)}$ specific to an extant sequence $S_j$, where $j \in \mathbb{J}$. *path*($\mathbf{E}^{(j)}$) recovers the indices in $\mathbb{I}^{(j)}$; $\mathbf{E}^{(j)}$ forms a single path of "character" nodes $X_{ji} = S_{ji}$ where $i \in \mathbb{I}^{(j)}$.

- An "input POG", denoted $\mathbf{E}* = \sum_{j \in \mathbb{J}} \mathbf{E}^{(j)}$ represents the joint set of edges collected from extant sequences. The presence of an edge between $a$ and $b$ is indicated by $\mathbf{E}*(a, b) > 0$.

- An "ancestor POG", is inferred to have a set of edges $\mathbf{E}^{(k)}$ where $k \in \mathbb{K}$. It links a series of nodes $Y_{ki}$ where $i \in \mathbb{I}^{(k)}$; each node either identifies a character state, or defines a probability distribution over character states. The latter is referred to as a "distribution" node. Once the POG for ancestor $k$ is inferred, *path*($\mathbf{E}^{(k)}$) recovers its valid indices $\mathbb{I}^{(k)}$;
as discussed in the next section, we use the below-diagonal elements to represent the state of support for *bi-directional* edges.

## Inference of ancestral states, insertions, and deletions

The phylogenetic tree with a nominated root and the collection of extant POGs serve as input to inference. GRASP supports two types of inference:

- marginal reconstruction at a specified ancestral branch point in the phylogenetic tree; as a result of inference, the nominated ancestor POG will contain distribution nodes that represent the marginal distributions of character states.

- joint reconstruction of all ancestral branch points; all ancestor POGs will contain character nodes that represent the most likely character states.

**Inferring insertions and deletions at ancestor branch points.** $\mathbf{E}^{(k)}$ defines all possible paths that can form a valid sequence at the branch point $k$ and therefore determines if a character state needs to be inferred at any given position. This section describes how *bi-directional edge* encoding refers to $\mathbf{E}^{(k)}$ and recovers $\mathbb{I}^{(k)}$ (valid positions in ancestor $k$) by implication. The principle is illustrated by example in Fig 6.

GRASP enumerates all edges in $\mathbf{E}*$ and infers $\mathbf{E}^{(k)}$ at each ancestor $k$ by either *maximum parsimony* or *maximum likelihood*. This inference involves solving multiple independent optimisation problems, avoiding scalability issues of earlier approaches [12, 13]. GRASP assigns scores (for parsimony) or probabilities (for maximum likelihood) to (combinations of) edges leaving and edges entering each reference position $i$. In other words, the goal is to infer for each position in an ancestor the best left and right neighbours. In the simplest case, a position in an ancestor has a single left neighbour and single right neighbour and this resolves to a linear sequence. In cases where a position has more than two neighbours, we benefit from representing these alternative neighbours as paths in a POG. This means that each edge is in fact evaluated twice: once in the forward direction, once in the backward direction. To formally distinguish the two evaluations, we refer to $(a, b)$ and $(b, a)$, where $a$ and $b$ are topologically sorted position indices; in other words, the diagonal of the adjacency matrix distinguishes forward from backward edges.

# a) Input POG

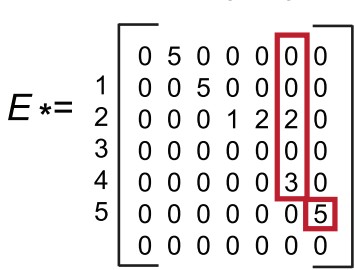

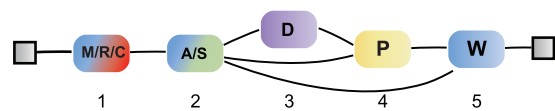

# b) Ancestor POGs

# c) Extant POGs

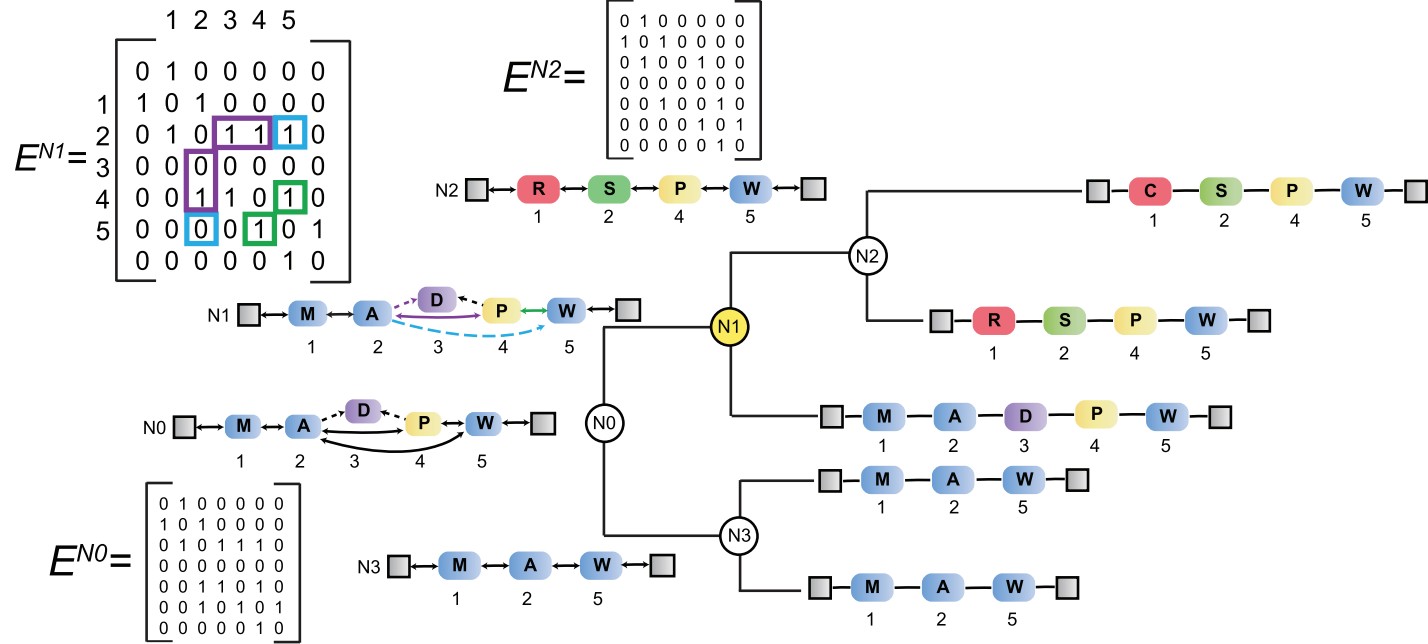

**Fig 6. A tree with five extant sequences and ancestors with bi-directional and uni-directional support for different edges shown. A**, Input POG and the matrix E* (recovering all edges under consideration for ancestors). Red squares refer to all edges involving position 5 (occupied by W in all sequences). Non-zero entries in the red squares identify indices for 2 and 4, as well as the end terminus. **B**, Ancestor POGs (labelled N0-N3) are shown at branch points as inferred by bi-directional edge parsimony. Solid arrows indicate bi-directional support; dashed arrows indicate uni-directional support. For uni-directional edges, the direction of support is shown by the direction of the arrow. The N1 ancestor is annotated to illustrate the relationship between indices in an ancestral matrix and the edges in the resulting ancestor POGs. The $(a, b)$ and $(b, a)$ indices are highlighted for select positions—for example, (4,5) and (5,4) are the forwards support from index 4—5 and backwards support from index 5—4 (green boxes and green edge in N1) showing that the edge from 4—5 is supported in both directions. The edge from 2—5 is supported going forward but not backward (blue boxes and blue edge in N1). Other edges forward from index 2 are maximally parsimonious in the forward direction at N1, such as 2—3 and 2—4, but only the support for 4 is reciprocated (purple boxes and purple edges in N1) recovering a "preferred" path from 2 to 4 to 5 in N1. **C**, Extant POGs, with position indices.

*Bi-directional edge parsimony* finds the edges at all ancestors that imply the lowest cost *across* the tree; the total cost is seen at the root node ($k = M + 1$). To trace the changes implied by the cost at the root, i.e. to determine what edge is preferred at each branch point, Eq 3 defines a score $\sigma_k^{next}(a, b)$ for each edge (*forward* from $a$ to $b$) at an ancestor $k$, which in turn depends on its children, recursively. That edge is scored for the same ancestor a second time

$\sigma_k^{prev}(b, a)$ (*backward* from $b$ to $a$) ([Eq 4](); [Fig 6]()). For each position and direction, the edges to choose between are available in $\mathbf{E}^*$; this number varies widely between positions and the direction considered.

As per standard parsimony, traversing the branch relative to an ancestor, the score of an edge depends on what edges are selected in the descendant branch points: retaining the edge is cost-free, whereas changing to a different edge will cost 1 ([5]()). The recursion ends at each extant sequence, at which a choice is made between an edge that either exists (as indicated by $\mathbf{E}^{(j)}(a, b) = 1$) or does not exist ($\mathbf{E}^{(j)}(a, b) = 0$). Taking the negative logarithm of this gives either 0 or $\infty$, forcing the selection of the edge extracted from the extant sequence (Eqs [3]() and [4]()). Edge choices that lead to the best parsimony score at the top-most branch point are traced back and recorded at each branch point $k$ in $\mathbf{E}^{(k)}$; considering direction, $\mathbf{E}^{(k)}(a, b) = 1$ if $(a, b)$ forms part of the optimal trace; likewise, $\mathbf{E}^{(k)}(b, a) = 1$ if $(b, a)$ is chosen, utilising either side of the diagonal of the matrix. For an ancestor $k$, its POG includes an edge $(a, b)$ if $\mathbf{E}^{(k)}(a, b) + \mathbf{E}^{(k)}(b, a) > 0$; if the sum is 2, that edge is *bi-directionally parsimonious*, implying preference when identifying ancestor sequences ([Fig 6]()).

$$\mathbf{E}^{(M+1)}\left(i, \arg\min_{b \in next(\mathbf{E}*, i)} \sigma_{M+1}^{next}(i, b)\right) = 1 \tag{1}$$

$$\mathbf{E}^{(M+1)}\left(\arg\min_{a \in prev(\mathbf{E}*, i)} \sigma_{M+1}^{prev}(i, a), i\right) = 1 \tag{2}$$

$$\sigma_k^{next}(a, b) = \sum_c^{\mathbb{Z}^{(k)}} \min_{i \in next(\mathbf{E}*, a)} \Delta(b, i) + \begin{cases} \sigma_c^{next}(a, i) & \text{if } c \in \mathbb{K} \ (c \text{ is ancestor}) \\ -\ln \mathbf{E}^{(c)}(a, b) & \text{otherwise } (c \text{ is extant}) \end{cases} \tag{3}$$

$$\sigma_k^{prev}(b, a) = \sum_c^{\mathbb{Z}^{(k)}} \min_{i \in prev(\mathbf{E}*, b)} \Delta(a, i) + \begin{cases} \sigma_c^{prev}(b, i) & \text{if } c \in \mathbb{K} \ (c \text{ is ancestor}) \\ -\ln \mathbf{E}^{(c)}(a, b) & \text{otherwise } (c \text{ is extant}) \end{cases} \tag{4}$$

$$\Delta(i, i') = \begin{cases} 0 & \text{if } i = i' \ (\text{no change}) \\ 1 & \text{otherwise } (\text{different edge}) \end{cases} \tag{5}$$

The equations define the implementation of bi-directional edge parsimony in GRASP, but one special case warrants attention; when one or more extant sequences is missing the reference position ($a$ in [Eq 3]() or $b$ in [Eq 4]()) a "no edge" state can be added to the total set of edges considered and thus compete with normal edges at ancestors. This option is currently available through the command line implementation of GRASP. To enhance readability, we have refrained from including it in the equations and matrices. The legacy approach (used for all experimentally validated predictions in this manuscript) is to (a) exclude the reference position at ancestors for which *all* extants are missing, and (b) infer a value for all remaining ancestors selected from those seen in other extants.

For *maximum likelihood*, like parsimony, each edge is evaluated twice, for each ancestor. Again, like parsimony, this is framed as two inferences for each reference position $i$: a choice between all edges leading forward from $i$ and a choice between all edges backward from $i$. Given that the number of edges that are considered jointly varies, we apply uniform evolutionary models with a congruent number of states, following the principle of Jukes-Cantor. Inference determines the combination of states, across all ancestors, that has the greatest joint probability, following the maximum likelihood approach described in the next subsection for

character states, which are similarly set at the leaves of the tree. As with parsimony, we need to consider the special case that arises from extant sequence that exclude characters at the reference position; again, a special "no edge" option competes with the edges that are indeed found in other extants at the same position. We note also that edges are attached to multiple positions and therefore branch lengths dictate how quickly they change, even when position-specific rates are available.

Both maximum parsimony and maximum likelihood optimise the selection of edges that have, independently, the same source or the same target, so there is no guarantee that there is a complete path through the POG where all edges are bi-directionally supported; however, in practice this turns out to be mostly the case.

**Inferring the character state of ancestor nodes.** For GRASP to infer character states and operate efficiently, we make several standard assumptions. First, we assume that each sequence position, $i$, can be modelled independently [40]. Second, we assume that character substitutions depend only on the state of the immediate ancestor [40].

GRASP itself does not estimate position-specific evolutionary rates [42, 43] but can use rates that are inferred jointly with the phylogenetic tree inference by tools such as IQ-TREE 2 [44]. In the absence of position-specific rates, GRASP assumes a fixed rate and that the tree is consistent with that.

The topology of the phylogenetic tree maps to a character tree for each position, subject to the position $i \in \mathbb{I}^{(k)}$ in an ancestor $k$ forming part of a valid sequence; for later, we define a transposition of that mapping as $k \in \mathbb{K}^{(i)}$, i.e. the subset of ancestors that have character content for a position $i$.

Each position-specific character tree maps to a directed Bayesian network, which is parameterised to reflect evolutionary distances (additively) at each branch, from the provided phylogenetic tree. The network is created with "observable" variables, instantiated to the characters in extant sequences $X_{ji} = x$. "Non-observable" variables in the Bayesian network correspond to the ancestors $Y_{ki}$ where $i \in \mathbb{I}^{(k)}$; how the character state or distribution for $Y_{ki}$ is inferred is described below.

A Bayesian network node is a conditional probability $P(X_{ji}|Y_{\kappa(j)i}, d_j)$ or $P(Y_{ki}|Y_{\kappa(k)i}, d_k)$, for $j \in \mathbb{J}^{(i)}$ and $k \in \mathbb{K}^{(i)}$ and is parameterised by their respective distances ($d_j$ or $d_k$; which refer to their closest ancestor branch point, $\kappa(j)$ or $\kappa(k)$, respectively).

The matrix of conditional probabilities is $e^{Q(d)}$ where $Q$ is the instantaneous rate matrix given by the evolutionary model. GRASP supports all popular models [45–48]. Inference of the joint ancestral character state at a position $i$ is then defined by:

$$
\begin{aligned}
P(\{Y_{ki} : k \in \mathbb{K}^{(i)}\}|\{X_{ji} : j \in \mathbb{J}^{(i)}\}, T) \propto \\
\prod_{j \in \mathbb{J}^{(i)}} P(X_{ji}|Y_{\kappa(j)i}, d_j) \prod_{k \in \mathbb{K}^{(i)}} P(Y_{ki}|Y_{\kappa(k)i}, d_k)
\end{aligned}
\tag{6}
$$

where $T$ is the tree with distances for all branches. The implementation uses an adaptation of variable elimination [22, 49], which decomposes the inference into an efficient series of products, given the hierarchical topology of the tree. Ancestral states are determined by the highest *joint probability* across all non-observed variables (all ancestors, all positions). From the above, GRASP is also capable of inferring the *marginal probability distribution* for each position in a given ancestor, by summing out all other non-observed variables. All inferences are exact (not approximated). GRASP can make use of position-specific, relative evolutionary rates, which are estimated by external tools.

Finally, GRASP facilitates an approach of exploring plausible alternative amino acids at sites with high Shannon entropy, by selecting residues that show a relatively high posterior probability in a marginal reconstruction [8]. Mutations can be introduced at these positions to test the robustness of predictions and to create alternative ancestors. GRASP extends this method, providing the ability to prioritise mutations that best capture inferred probability distributions by minimising the expected relative entropy.

### Identifying a single, preferred ancestor sequence

Not uncommonly, multiple indel histories are equally parsimonious or the preferred edge in one direction is not preferred in the other, implying that several ancestor candidate sequences can be identified by traversing an ancestor POG; however, in some applications it is necessary to nominate a single sequence.

To determine a "preferred" path through an ancestor POG, we first define a subset of extant sequences $\mathbb{J}^{(k)}$ that are in the subtree under a given ancestor, $k$. To express preference between multiple edges, we calculate the proportion of extant sequences that contain a particular edge (see Eq 7).

$$w_k(a, b) = \frac{\sum_{j \in \mathbb{J}^{(k)}} \begin{cases} 1 & \text{if } \mathbb{E}^{(j)}(a, b) = 1 \\ 0 & \text{otherwise} \end{cases}}{|\mathbb{J}^{(k)}|} \tag{7}$$

**Identifying the preferred path.** Like a Markov chain, GRASP uses weights (Eq 7) to represent transition probabilities and can use these to find the most probable path, enforcing a preference for bi-directional edges.

As an alternate option, GRASP also implements the A* algorithm [50] to determine the selection of edges in a POG that jointly minimises the cost, travelling from the N- to the C-terminus. Here, the cost assigned to an edge is given by Eq 8.

$$\gamma_k(a, b) = (1 + (\eta_k(a, b) \cdot (1 - w_k(a, b)))) \cdot (b - a) \tag{8}$$

$\eta$ is defined in Eq 9 and imposes a preference for bi-directional edges; a uni-directional edge is only chosen in the absence of bi-directional edges to complete the traversal. The exception is the edge to the first node, and the edge from the last node, where bi-directionality is disregarded. The impact of the weight is normalised by the number of positions skipped by a given edge, $b - a$. This ensures that each complete ancestral sequence is scored evenly, regardless of the number of edges it takes to form.

$$\eta_k(a, b) = \begin{cases} N & \text{if } \mathbf{E}^{(k)}(a, b) + \mathbf{E}^{(k)}(b, a) < 2, \ a \neq 0 \text{ and } b \neq N + 1 \\ 1 & \text{otherwise} \end{cases} \tag{9}$$

### GRASP, FastML, and PAML comparison method

The following procedure was used to evaluate each of the tools: (1) the input multiple sequence alignment was randomly divided into $G$ groups of alignments with approximately equal numbers of sequences, where $G \in \{5, 10, 20\}$; (2) for each sub-alignment, the input phylogenetic tree constructed from the full alignment was pruned to remove sequences not in the sub-alignment. To represent the same principal ancestor across all groups, as well as to maintain a valid tree, branches in the original phylogenetic tree with removed sequences were collapsed and

branch distances added together; (3) sub-alignments and corresponding pruned trees were therefore pared-down representatives of the same family and used as input to each of the ASR tools; (4) the process was repeated until 20 ancestral sequences had been generated for each configuration, e.g., when $G = 5$ the process is repeated four times. This procedure therefore results in sequences that belong to multiple groups across replicates for $G = 5$ and $G = 10$.

The JTT evolutionary rate model [46] was used for all inferences and variable rates were calculated from a discrete gamma distribution with eight categories. To remove confounding effects of different strategies for dealing with gaps, we removed any column that contained a deletion, leaving 455 and 242 columns in the KARI and CYP2 multiple sequence alignments, respectively.

## Indel inference comparison method

INDELible was used to generate indel histories that followed a Zipfian distribution with an alpha value of 1.7 [51], maximum indel length of 10, LG substitution model, with all trees set to have an evolutionary depth of 1 and an original ancestral sequence length of 300 amino across four indel rates (relative to substitution rate) of 0.001, 0.005, 0.01, and 0.03 across four taxon sizes of 100, 250, 500, and 750 extant sequences. Any entirely empty columns in the alignment of extants (i.e., where there was character content in an ancestor that was completely absent across all extant sequences) were removed from both the extant and ancestral alignments.

The known alignments and trees were provided to GRASP and each of the indel inference methods used to predict a set of ancestral sequences. The ancestral alignments from each method as well as the true history as generated by INDELible were used to generate sets of indels defined by the specific branch and start and end position of the indel event.

The unaligned extant sequences were then also aligned by MAFFT (FFT-NS-2) and the analysis re-performed with realigned alignments and known trees provided to GRASP.

## Tool availability

GRASP is freely accessible via a web server at http://grasp.scmb.uq.edu.au. The online service allows users to upload their own data sets and predict ancestors. The results are presented to allow exploration of ancestral POGs and their states via an interactive phylogenetic tree. Numerous other functions are available including annotation of trees with taxonomy and user specified terms, inspection of probability distributions for the identification of mutations for alternative ancestors, and sharing of entire reconstructions. A tutorial, user guide, and several example reconstructions are also available from the web site.

Additional information and a suite of tools to assist in the application of GRASP are available at https://bodenlab.github.io/GRASP-suite. An implementation of bi-directional edge parsimony indel inference that considers gaps as missing data rather than explicitly encoding gaps as an additional "no edge" state is available through the web server. All other indel inference methods as well as the option to either consider gaps as missing data or additional states are available through a separate command line version. The implementation in Java and a web application are available from the same site. The software is available under the GNU General Public License v3.0.

## GDH-GOx experimental methods

### GDH-GOx ancestral inference

Starting from an aligned data set and phylogenetic tree previously established by Sützl et al. [18] for the GDH-GOx cluster, only the four major clades (GOx, GDH I, GDH II, and GDH

III) were selected, together with the second small GDH III clade. All sequences with >800 amino acids as well as manually selected sequences showing large insertions were removed from the selection, resulting in 399 sequences. This sequence selection was aligned by MAFFT v7.271 G-INS-i [52] and the alignment trimmed for positions with >99% gaps by trimAl v1.2 [53]. The alignment was pruned using Gblocks 0.91b [54] with a less stringent block selection and the phylogenetic tree was inferred by PhyML [55] using default settings, with subtree pruning and regrafting to optimise tree topology, smart model selection, and aLRT SH-like branch support calculated. The tree was rooted on the midpoint. The full alignment, trimmed by trimAl and with a trimmed N- and C-termini but not pruned by Gblocks was used to perform a marginal reconstruction of ancestral nodes with the LG evolutionary rate model [48], with indels inferred using bi-directional edge parsimony.

## GDH-GOx synthesis and cloning

The N- and C-terminal sequences not present in the ancestral sequences were replaced by the equivalent amino acid sequences of GOx from *Aspergillus niger*, MQTLLVSSLVVSLAAAL PHYIRSNGIEASLLTDPKDVSGRT and ASMQ, respectively. Resulting ancestral genes were codon-optimised for *Komagataella phaffii* (formerly *Pichia pastoris*) expression, ordered from BioCat GmbH, and cloned into the expression vector pPICZ A together with an added poly-histidine tag (6 x His). Constructs were linearized with *Pvu*II and transformed into *K. phaffiii* via electroporation.

## GDH-GOx expression

Ancestral and extant GOx and GDH genes were expressed in *K. phaffiii* under the AOX1 promoter with methanol induction. Routine cultivations and selection of the transformed cells were done in liquid YPD medium supplemented with zeocin (100 mg/L) at 30˚C and 130 rpm. Expression was done in shake flasks at 30˚C and 130 rpm on modified BMMY medium (20 g/L peptone from casein, 10 g/L yeast extract, 100 mM potassium phosphate buffer pH 6.0, 10 g/L $(NH_4)_2SO_4$, 3.4 g/L yeast nitrogen base (without amino acids and $(NH_4)_2SO_4$), and 0.4 mg/L biotin) containing 12 g/L sorbitol and 2% methanol. After centrifugation at 6,000 x *g* and 4˚C for 30 minutes, supernatants were loaded onto a 5 mL HisTrap column (GE Healthcare) equilibrated with binding buffer (50 mM potassium phosphate buffer pH 6.5, 500 mM NaCl, and 20 mM imidazole) and washed for at least 10 column volumes. Proteins were eluted using a linear gradient from 20 to 500 mM imidazole (50 mM potassium phosphate buffer pH 6.5, 500 mM NaCl and 500 mM imidazole). Manually collected fractions were concentrated and desalted (50 mM phosphate buffer pH 6.5) in Vivaspin 20 tubes (Sartorius) with 30,000 Da molecular mass cut-off.

## GDH-GOx activity assays

Both GDH and GOx activity were measured spectrophotometrically at 30˚C on a UV/Vis spectrophotometer (Lambda 35, Perkin Elmer), using appropriately diluted enzyme solution, 20 mM D-glucose, and the respective electron acceptor in 50 mM potassium phosphate buffer pH 6.5. The electron acceptors 1,4-benzoquinone (BQ) and ferrocenium-hexafluorophosphate (FcPF6) were used at 0.5 and 0.2 mM, and their reduction was followed at 290 and 300 nm, respectively. Reduction of the electron acceptor oxygen was measured using the peroxidase-coupled 2,2'-azino-bis(3-ethylbenzothiazoline-6-sulphonic acid) (ABTS) assay [56], following the reduction of 0.1 mM ABTS at 420 nm.

### GDH-GOx thermal stability assays

The thermal stability of GDH-GOx enzymes was assessed by differential scanning calorimetry conducted on a PEAQ-DSC automated instrument (Malvern Panalytical). All enzyme samples were diluted to 5 μM (∼ 0.33 mg/ml) in 50 mM potassium phosphate buffer pH 6.5, and scanned from 2090˚C with a scan rate of 60˚C/h and feedback set to high. Instrument blanks were recorded using buffer only and rescans were measured for all samples. Data analysis was performed using the MicroCal PEAQ-DSC software V.1.22. The background signal was subtracted using rescans whenever applicable or buffer blanks otherwise, the baseline was fitted using the spline method, and peaks were fitted with a non-two-state model.

## DHAD experimental methods

### DHAD ancestral inference

A *minimal* 585-sequence set was created, consisting of members annotated with family Ilvd/Edd" in UniProt, and excluding sequence fragments (as defined by UniProt). Most sequences were from Swiss-Prot, with several TrEMBL entries added due to function and structural data being available. A *baseline* data set of 1,612 sequences was created from the minimal data set, ensuring that 19 nominated enzymes with experimental data (functional and/or structural) were included, as well as members of their UniRef90 clusters. A *background* data set of 8,221 sequences included annotated members from the broadest assortment of species, only filtered to be non-redundant at 90% identity (using UniRef90). All three data sets were checked for the aligned location of two motifs (CDK and PCN/PGH/SAH with provision for a substitution) that are associated with the active site [57]; sequences that did not exhibit these motifs were removed.

The background data set was then repeatedly and independently sampled to extend the baseline data set to up to 9,112 sequences. At each size increment of 500 sequences an alignment was created using Clustal Omega [58], and a phylogenetic tree was inferred using FastTree [59] and rooted using phosphogluconate dehydratase as an outgroup. Despite differences in alignments and phylogenetic trees at each data size increment, we were able to map any ancestor in a smaller tree to an ancestor in a larger tree by maximising shared inclusions and exclusions of member proteins of the ancestral subtrees. Joint reconstruction was performed with indels inferred with bi-directional edge parsimony and the JTT evolutionary model [46].

### DHAD synthesis and cloning

The inferred DHAD ancestral genes N1, N423, and N560 from the 585 data set, and the equivalent nodes N9, N1442, and N1443 from the 9,112 data set were optimised for *E. coli* expression and synthesised by Twist Bioscience and ATG:biosynthetics GmbH, respectively. After amplification, the purified DNA fragments were digested with *Sap*I, then ligated into a modified pET26 vector (p7XNH3) [60].

### DHAD expression

Expression of the DHAD genes was performed in shaking flasks. *E. coli* BL21 (DE3) cells transformed with the p7XNH3 plasmid and the appropriate inserted gene fragment were grown as an overnight pre-culture in lysogeny broth supplemented with kanamycin (100 μg/ml), and then inoculated 1:50 into auto-induction ZP-5052 medium [61] supplemented with kanamycin (100 μg/ml). These cultures were incubated at 90 rpm and 37˚C for 3 hours and then overnight at 18˚C in a horizontal orbital shaking incubator. Cells were disrupted by sonication in

binding buffer (50 mM potassium phosphate buffer, 500 mM NaCl, 10% glycerol, and 20 mM imidazole) at pH 8.0. Cell debris was pelleted by centrifugation. Proteins were purified using an ÄKTA Purifier FPLC system and a HisTrap HP Nickel column (GE Healthcare). Filtered samples were loaded onto the column and washed with binding buffer. The His-tagged proteins were then eluted with elution buffer (50 mM potassium phosphate buffer, 500 mM NaCl, 10% glycerol, and 500 mM imidazole) at pH 8.0. Desalting of the enzymes was carried out using HEPES buffer pH 7.0.

## DHAD activity assays

DHAD activity was analysed by HPLC of an assay mixture containing the respective DHAD, 25 mM HEPES buffer pH 7.0, 5 mM MgSO$_4$, and 25 mM of sodium D-gluconate, and incubated at 30˚C. Samples were taken every few hours for 3 days. The enzyme was removed by ultrafiltration (PES 10 kDa MWCO, VWR) and the samples were stored at -20˚C until analysed by HPLC. HPLC measurements were performed on an Ultimate-3000 HPLC system (Dionex), equipped with an auto-sampler and diode-array detector. D-gluconate and products were separated by using a Metrosep A supp10–250/40 column (250 mm, particle size 4.6 μm, Metrohm) at 65˚C by isocratic elution with 12 mM ammonium bicarbonate at pH 10, followed by a washing step with 30 mM sodium carbonate at pH 10.4 and a flow rate of 0.2 ml/min. Each sample injection volume was 10 μl. System peak calibration was performed using external standards of the known compounds.

## CYP2U experimental methods

### CYP2U ancestral inference

Five candidate CYP2U proteins were chosen, one each from *Andrias davidianus, Python bivittatus, Marmota marmota, Poecilia reticulata*, and *Amazona aestiva*. A BLASTp search of each of the candidates was conducted, excluding hits from plants (taxonomic id:3193) or fungi (taxonomic id:4751) (E-Value = 0.00001). Sequences from the BLASTp search were retained if they had at least 55% sequence identity to the original candidate sequence. This procedure was also repeated retaining sequences with at least 50% sequence identity, however, the additional sequences from this lower bound were all removed at later stages of curation, indicating that 55% was an appropriate level of identity to identify CYP2U subfamily sequences. Sequences from the BLASTp searches were collated and duplicate, identical sequences were removed. Manual curation was performed to remove sequences below 400 amino acids in length or containing unidentified amino acids. Sequences were aligned using MAFFT (L-INS-i) with default parameters [52]. Removal of sequences with indel events over 20 amino acids (suggestive of incorrect annotation of splice sites) was completed in an iterative manner by first identifying which sequence had the longest indel over 20 amino acids, removing it and realigning the remaining sequences, and then continuing until no sequence had an indel over 20 amino acids. Sequences were manually inspected and any sequences with apparent frameshift mutations were removed. Sequences were mapped back to their exon structure and removed if they had more than two exons difference to the accepted number of five exons for CYP2U sequences, indicative of poor sequence quality or poor sequence annotation.

Sequences missing the conserved cysteine residue characteristic of cytochrome P450 enzymes were removed. Similar procedures were used to generate the CYP2R and CYP2D subfamily alignments, then the full set of all three subfamilies was combined and realigned in MAFFT to generate the largest data set. Phylogenetic trees were inferred using RAxML [62]. A CYP2R *Latimeria chalumnae* sequence (XP_005989762.1) was manually shifted on the phylogenetic tree to better represent the known phylogeny [63], while retaining each sequence's

overall evolutionary distance to the root. Joint reconstruction was performed with indels inferred with bi-directional edge parsimony and the JTT evolutionary rate model [46].

## CYP2U synthesis and cloning

The amino acid sequences of CYP2U ancestors were inferred starting from the conserved PPGP motif, which signifies the junction between the transmembrane anchor and the catalytic domain. For expression of the resurrected ancestors in bacteria, this region was replaced with an N-terminal sequence (MAKKTSSKGKL), which is known to improve expression yields of microsomal P450s in bacteria [64] and had been used to express human CYP2U1 in *E. coli* [65]. To enable purification, a flexible ST linker followed by a polyhistidine tag (6 x His) was added to the C-termini of the sequences. All ancestor sequences were codon-optimised for *E. coli* expression, and the N-termini were optimised initially using mRNA optimiser [66] and subsequently manually until the predicted free energy of secondary structure formation was greater than -15 kJ/mol. The genes were synthesised as GeneStrings (GeneArt, Invitrogen) designed with 60 bp 5 and 3 end extensions complementary to the pCW vector, cloned by Gibson assembly, and then sequence-verified by dideoxy sequencing (Australian Genome Research Facility). Correct inserts were subcloned into a bicistronic pCW vector upstream of the open reading frame for the human cytochrome P450 reductase (hCPR) using the *Nde*I and *Xba*I sites.

## CYP2U expression

DH5$\alpha$ F IQ *E. coli* cells carrying the pGro7 plasmid were transformed with pCW vectors containing the relevant P450 and CPR genes or the empty vector (pCW controls"), and selected using chloramphenicol (20 g/ml) and ampicillin (100 g/ml). Single colonies were used to inoculate overnight cultures in lysogeny broth with antibiotics, as above. Batch cultures were grown at 25˚C, 180 rpm in 500 ml flasks containing 50 ml terrific broth supplemented with trace elements, 1 mM thiamine, and antibiotics. Cultures were induced after 5 hours with 1 mM IPTG and 4 mg/ml L-arabinose, and supplemented with 500 mM delta-aminolaevulinic acid. Cultures were grown for a further 43 hours before harvesting by centrifugation at 6,000 x *g* for 10 minutes. *E. coli* pellets were weighed and resuspended in 2 ml/g (wet weight) sonication buffer (100 mM potassium phosphate buffer pH 7.4, containing 20% (w/v) glycerol, 6 mM magnesium acetate, 1 mM PMSF, and a protease inhibitor cocktail (Sigma-Aldrich)). Cells were lysed using a Constant Systems OneShot cell disruptor followed by centrifugation at 10,000 x *g* for 20 minutes. The supernatant was centrifuged at 180,000 x *g* for 1 hour and the pellet was resuspended in TES buffer (100 mM Tris acetate, 500 mM sucrose, and 0.5 mM EDTA pH 7.6) using a Potter-Elvehjem homogeniser. The P450 concentration was determined in intact cells and membranes using Fe(II).CO vs. Fe(II) difference spectroscopy [67].

## CYP2U activity assays

P450 (0.02 $\mu$M), added in membranes prepared from bacteria coexpressing hCPR, was premixed with 50 $\mu$M luciferin CEE or luciferin ME-EGE (Promega) in 100 mM potassium phosphate pH 7.4, and incubated at 37˚C for 10 minutes. Reactions were initiated by addition of the NADPH-regenerating system (NGS; 0.25 mM NADP$^+$, 10 mM glucose-6-phosphate, and 0.5 U/ml glucose-6-phosphate dehydrogenase), and incubated with gentle shaking at 37˚C for 30 minutes. An equal volume of the luciferin detection reagent was added, and reactions were incubated for a further 20 minutes at room temperature. Luminescence was measured using a CLARIOstar multimodal plate reader (BMG Labtech).

### CYP2U thermal stability assays

Ancestors were expressed in *E. coli* as described above and cell pellets were resuspended in whole cell spectral assay buffer (WCAB; 100 mM potassium phosphate, 20 mM D-glucose, and 6 mM magnesium acetate pH 7.4) to one eighth of the original culture volume. The resuspended cultures, distributed into tubes in 200 $\mu$L volumes, were incubated at a range of temperatures (25–80˚C, in 5˚C increments) for 60 minutes, followed by a 5 minute recovery at 4˚C and equilibration at 25˚C. The remaining P450 content was measured in intact cells using the method of Johnston et al. [67]. The proportion of total P450 content compared to the unheated control (25˚C) was plotted against temperature and the $^{60}T_{50}$ value was calculated by fitting the data to a variable slope (4-parameter) dose response curve in GraphPad Prism 8.0.

## KARI experimental methods

### KARI ancestral inference

We created two separate data sets representing KARI class I and class II, respectively. The sequence alignment for class II was taken directly from Gumulya et al. [8] and used to compare tools. Class I sequences were compiled by searching for both reviewed and unreviewed proteins in UniProt, designated as bacterial and belonging to the family (26,485 sequences). We removed all fragments and sequences above the length of 400 to exclude obvious cases of class II enzymes. The sequence set was redundancy-reduced with CD-HIT at 99% [68], resulting in 11,920 sequences, from which 57 sequences were manually removed by observing a C-terminal knotted domain, indicative of class II. After aligning all sequences with Clustal Omega [58], several sequences appeared poorly aligned, mostly due to unique but substantial truncation or elongation.

To objectively identify such outliers, we determined the proportion of gaps in each column (number of sequences with gaps in given column / number of sequences), and used this to calculate the Shannon entropy for a given sequence at each position (entropy of either gap or amino acid, as found at the position). The position-specific entropies were then summed across each aligned sequence, for all sequences. This demarcated approximately the top 1% of sequences due to their atypical gap patterns, which we decided to remove, resulting in a final set of 11,756 sequences.

Phylogenetic tree inference was carried out using FastTree [59].

The KARI class I data sets were created by decreasing their size from 11,756 via 10 regular decrements reaching a minimum representation of 1,176 sequences. For each subset, sequences were randomly removed and the alignment was recalculated independently. For each alignment, a new tree was calculated, and rooted using KARI sequences in Aquificae and Thermotogae as an outgroup. For each subset, we computed reconstructions for 50 randomly chosen ancestor nodes (mapped across each tree as described for the DHAD data sets). Joint reconstruction was performed with indels inferred with bi-directional edge parsimony and the JTT evolutionary model [46].

## Supporting information

**S1 Table. Comparison of thermal transitions.** Differential scanning calorimetry of an extant glucose oxidase from *Aspergillus niger*, the ancestor inferred at node N320, and the ancestor inferred at node N320 with a single amino acid change, based on marginal distributions. (XLSX)

**S1 Fig. Percentages of gaps in sequence alignments.** Percentage of positions across every sequence in a given alignment that contains a gap character, for the nine data sets, the

simulated data from INDELible, and the simulated data from INDELible as realigned by MAFFT. Simulated data is shown for 500 extant sequences at indel rates of 0.001, 0.005, 0.01, and 0.03.
(EPS)

**S2 Fig. Distributions of contiguous gap lengths within alignments. A**, CYP2U, DHAD, KARI, and GDH-GOx datasets. **B**, The simulated data from INDELible. **C**, The simulated data from INDELible as realigned by MAFFT. Simulated data is shown for 500 extant sequences at indel rates of 0.001, 0.005, 0.01, and 0.03. Note that truncations of sequence content are also counted in these distributions.
(EPS)

**S3 Fig. Overview images of the alignments generated for the four indel rates used in the simulated indel evaluation.** Each alignment is the true alignment simulated by INDELible for 250 extant sequences with an indel rate of either 0.001, 0.005, 0.01, or 0.03.
(EPS)

**S4 Fig. Evaluation of indels with realigned data. A**, Root length generated from realigned data at four taxon sizes at an indel rate of 0.03 (n = 5). **B**, Venn diagram showing the overlap of specific indels identified by each method at taxon size 750 and an indel rate of 0.03. Total numbers of indels are shown in the largest Venn diagram and subsets of this data according to indel type and indel size are shown in the smaller Venn diagrams. Not all intersections are shown ($N = 1$).
(EPS)

**S5 Fig. Number of indels uniquely identified by each indel method.** At four taxon sizes at two indel rates (0.005 and 0.01), organised by indel type and size ($N = 5$). Note the change in range of the y-axis between indel rates.
(EPS)

**S6 Fig. Expression of CYP2U hybrid ancestors. A**, Fe(II) vs. Fe(II).CO difference spectra for CYP2U ancestors in *E. coli* membranes. **B**, Expression level of CYP2U ancestors in *E. coli* cultures, quantified using Fe(II) vs. Fe(II).CO difference spectroscopy. Data are means +/- SEM, $N = 3$.
(EPS)

**S7 Fig. AlphaFold2 models of CYP2U ancestral and extant proteins. A**, AlphaFold2 model of human CYP2U1. The AA' loop is shown in red, the D-E loop is shown in blue, and the haem cofactor is shown in pink. **B**, Structural alignment of the AlphaFold2 models for human CYP2U1 (beige) and CYP2U_N1 (black) with the human PDB structure for CYP2R1 (green), which is the closest relative of CYP2U1 with an experimental structure. CYP2R1, like all other CYP2s, lacks the AA' loop extension found in CYP2U proteins. The haem cofactor is depicted in red. **C**, AlphaFold2 structure for human CYP2U1 showing the low model confidence for the AA' loop extension. **D**, Structural alignment of the AlphaFold2 models for all the CYP2U ancestors and three extant CYP2U proteins (human CYP2U1, mouse CYP2U1, zebrafish CYP2U1), showing the differences in the predicted positioning of the AA' loop extension. The haem cofactor is shown in red. **E**, Structural alignment of the AlphaFold2 models for the CYP2U_N1 ancestors with and without the AA' loop insertion. Aside from the AA' loop, no substantial structural differences are observed between these structures. **F**, Structural alignment of the AlphaFold2 models for all the CYP2U ancestors and three extant CYP2U proteins (human CYP2U1, mouse CYP2U1, zebrafish CYP2U1), showing the elongation of the D-helix

and D-E loop in the zebrafish CYP2U1 (blue) and CYP2U_N5 (grey) with LSEE insertion.
(EPS)

**S8 Fig. Predicted ancestor sequence lengths are unaffected by size of reconstruction.** Mean
and standard deviation of the lengths of 50 ancestor sequences mapped are plotted for different
reconstructions and data set sizes for DHAD and KARI.
(EPS)

**S9 Fig. Run times for the DHAD and KARI enzyme families as data set size increases.**
Reconstructions were performed using GRASP running on 64 GB RAM, 5 threads on 2x 2.6
GHz 14C Xeon VM.
(EPS)

**S10 Fig. Thermal shift assays for DHAD ancestors.** There is an increase in temperature
between equivalent ancestral nodes N423 (585 data set size) and N1442 (9,112 data set size),
and equivalent ancestral nodes N560 (585 data set size) and N1443 (9,112 data set size).
(EPS)

**S11 Fig. Thermal stability and activity for the CYP2U, CYP2U/CYP2R, and CYP2U/
CYP2R/CYP2D ancestors with luciferin MultiCYP. A**, Comparison of $T_{50}$ values after a 60
minute incubation at a range of temperatures (25–80˚C). Data are means +/- SEM, $N$ = 2. **B**,
Turnover of luciferin MultiCYP by CYP2U ancestors in *E. coli* membranes, also containing
human CPR, after 30 minutes at 37˚C. Membranes from cells expressing only human CPR are
included as a negative control. Data are means +/- SEM, $N$ = 3. The two graphs represent two
independent experiments with two independent levels of human CPR activity recorded.
(EPS)

**S12 Fig. Comparison between ancestors generated using GRASP, FastML, and PAML. A**,
Expression level of CYP2U ancestors in *E. coli* cultures quantified using Fe(II) vs. Fe(II).CO
difference spectroscopy. Data are means +/- SEM, $N$ = 3. **B**, Fe(II) vs. Fe(II).CO difference
spectra for ancestors generated using GRASP, FastML, and PAML in *E. coli* membranes. **C**,
Turnover of luciferin MultiCYP by CYP2U ancestors in *E. coli* membranes, also containing
human CPR, after 30 minutes at 37˚C. Membranes from cells expressing only human CPR are
included as a negative control. Data are means +/- SEM, $N$ = 3. **D**, Comparison of $T_{50}$ values
after a 60 minute incubation at a range of temperatures (25–80˚C) for ancestors generated
using GRASP, FastML, and PAML. Data are means +/- SEM, $N$ = 2.
(EPS)

**S13 Fig. Tool comparison using CYP2 data. A**, Average fractional distance between tools,
calculated as pairwise fractional distances for each ancestral prediction for a given tool against
all other ancestral predictions of other tools at 5 groups of 195 sequences, 10 groups of 97 or 98
sequences, and 20 groups of 48 or 49 sequences. Parameter combinations are joint (J) and
marginal (M) reconstruction; and fixed or variable evolutionary rates (FastML and PAML
only). **B**, Average fractional distance between a better-sampled ancestor inferred by GRASP
using 975 sequences and each tool / parameter combination at 5, 10, and 20 groups.
(EPS)

**S14 Fig. Statistical evaluation of determinants of ancestor prediction performance using
1,682 KARI sequences. A**, Between tool distances grouped by tool within data set size. **B**, Dis-
tance to better-sampled ancestor grouped by tool within data set size. **C**, Distance to better-
sampled ancestor grouped by size within tool. **D**, Distance to better-sampled ancestor grouped
by rate parameter within data set size. PAML was excluded for the largest data set size; variable

rates were not used for the largest data set size. All p-values were determined by a two-tailed Student's $t$-test. Only significant comparisons are shown (* means $p < 0.05$, ** means $p < 0.01$, *** means $p < 0.001$, **** means at limits of precision of test). All parameter settings are from Fig 5.
(EPS)

**S15 Fig. Statistical evaluation of determinants of ancestor prediction performance using 975 CYP2 sequences. A**, Between tool distances grouped by tool within data set size. **B**, Distance to better-sampled ancestor grouped by tool within data set size. **C**, Distance to better-sampled ancestor grouped by size within tool. **D**, Distance to better-sampled ancestor grouped by rate parameter within data set size. All p-values were determined by a two-tailed Student's $t$-test. Only significant comparisons are shown (* means $p < 0.05$, ** means $p < 0.01$, *** means $p < 0.001$, **** means at limits of precision of test). All parameter settings are from S13 Fig.
(EPS)

**S16 Fig. Run times of GRASP, FastML, and PAML at different parameter combinations and group sizes on the KARI data set.** Parameter combinations are joint and marginal reconstruction; and fixed or variable evolutionary rates (FastML and PAML only).
(EPS)

**S17 Fig. Mean sequence identity between extant sequences and a selected ancestral node.** Positions to compare are identified from the multiple sequence alignment for each data set. The selected ancestral sequence is N0, the root node, from a joint reconstruction for each data set except for GDH-GOx, in which the marginal reconstruction of N320 is the selected ancestral sequence.
(EPS)

## Acknowledgments

We thank Broder Rühmann who helped prepare data for the manuscript.

## Author Contributions

**Conceptualization:** Gabriel Foley, Elizabeth M. J. Gillam, Mikael Bodén.

**Data curation:** Gabriel Foley, Leander Sützl, Mikael Bodén.

**Formal analysis:** Gabriel Foley, Mikael Bodén.

**Funding acquisition:** Bostjan Kobe, Luke Guddat, Gerhard Schenk, Burkhard Rost, Dietmar Haltrich, Volker Sieber, Elizabeth M. J. Gillam, Mikael Bodén.

**Investigation:** Bostjan Kobe, Ross T. Barnard, Luke Guddat, Gerhard Schenk, Burkhard Rost, Dietmar Haltrich, Volker Sieber, Elizabeth M. J. Gillam, Mikael Bodén.

**Methodology:** Gabriel Foley, Ariane Mora, Connie M. Ross, Scott Bottoms, Leander Sützl, Julian Zaugg, Raine E. S. Thomson, Yosephine Gumulya, Elizabeth M. J. Gillam, Mikael Bodén.

**Project administration:** Mikael Bodén.

**Resources:** Gabriel Foley, Mikael Bodén.

**Software:** Gabriel Foley, Ariane Mora, Marnie L. Lamprecht, Julian Zaugg, Alexandra Essebier, Brad Balderson, Rhys Newell, Mikael Bodén.

**Supervision:** Ross T. Barnard, Luke Guddat, Gerhard Schenk, Burkhard Rost, Dietmar Haltrich, Volker Sieber, Elizabeth M. J. Gillam, Mikael Bodén.

**Validation:** Connie M. Ross, Scott Bottoms, Leander Sützl, Raine E. S. Thomson, Jörg Carsten, Yosephine Gumulya, Dietmar Haltrich, Elizabeth M. J. Gillam.

**Writing – original draft:** Gabriel Foley, Connie M. Ross, Scott Bottoms, Leander Sützl, Jörg Carsten, Yosephine Gumulya, Elizabeth M. J. Gillam, Mikael Bodén.

**Writing – review & editing:** Gabriel Foley, Connie M. Ross, Leander Sützl, Bostjan Kobe, Ross T. Barnard, Luke Guddat, Gerhard Schenk, Burkhard Rost, Dietmar Haltrich, Volker Sieber, Elizabeth M. J. Gillam, Mikael Bodén.

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
