## [Decision Letter · Decision Letter 0]

14 Apr 2022

Dear Dr. Boden,

Thank you very much for submitting your manuscript "Engineering indel and substitution variants of diverse and ancient enzymes using Graphical Representation of Ancestral Sequence Predictions (GRASP)" for consideration at PLOS Computational Biology.

As with all papers reviewed by the journal, your manuscript was reviewed by members of the editorial board and by several independent reviewers. In light of the reviews (below this email), we would like to invite the resubmission of a significantly-revised version that takes into account the reviewers' comments.

Besides addressing all other comments from the reviewers, please make every effort to render the algorithm at the basis of your method clear and understandable. 

We cannot make any decision about publication until we have seen the revised manuscript and your response to the reviewers' comments. Your revised manuscript is also likely to be sent to reviewers for further evaluation.

Sincerely,

Marco Punta

Associate Editor

PLOS Computational Biology

Arne Elofsson

Deputy Editor

PLOS Computational Biology

Reviewer's Responses to Questions

**Comments to the Authors:**

Reviewer #1: This study aims to provide a new way to treat indel events for ancestral sequence reconstruction. This is an important problem for this field and better methods would be very welcome. Unfortunately I am in no position to evaluate this manuscript. I could not follow the mathematical descriptions of what this method does exactly. This is not meant as a judgment of the authors' work; this reviewer was simply to stupid to understand the method. I would suggest, however, that the authors describe the method in the main text and accompany it with figures that explain how it works. As it stands, the new method is only described in the methods section.

Reviewer #2: This paper introduces a program called GRASP, which performs Indel estimation in ancestral sequence estimation, shows its applications, and attempts to demonstrate the usefulness of this program. Ancestral sequence reconstructions have become widely used in the research fields of molecular evolutionary biology and protein engineering. In this context, the accurate estimation of indels is important for the creation of proteins that actually have the "correct" structure and functionality to be analyzed based on the estimated sequence.

　GRASP, presented in this paper, can infer ancestral sequences including Indel in six different ways, given an alignment and phylogenetic tree. It is important that the user can choose the method of analysis. In addition, the authors actually used this program to perform ancestral sequence estimation on data from different protein families, and actually produced some of the ancestral proteins for functional analysis. In addition, the authors have also performed functional analysis of proteins based on the ancestral sequence estimation results based on different data sizes (number of sequences in the alignment/phylogenetic tree) for each protein family. The results of these analyses suggest that GRASP's ancestral sequence estimation, including Indel, is relatively robust to differences in protein family types and analysis data sizes. This should be useful for researchers who perform many ancestral sequence reconstructions.

　However, we have the following questions and comments about this paper.

1 The authors have performed ancestral sequence reconstruction experiments, including actual ancestral protein production using GRASP, for three protein families (CYP2U/CYP2R/CYP2D, DHAD, and KARI). As the basal data, the extent to which the sequence data for each of the present organisms falls within the range of sequence similarity should be clearly indicated. In addition, the readers should be able to access the actual alignment, phylogenetic tree, and expressed and analyzed protein sequences. This will allow the readers to better judge the validity of the authors' analyses.

2 In relation to the above section, it is my understanding that only analyses that do not take into account differences in evolutionary rates between loci when estimating ancestral sequences are valid for GRASP. Note that I have also accessed the web page of the authors' web service. I wonder if such an analysis would be able to recover reasonable amino acid sequences and the presence or absence of indels, especially when using data with low sequence conservation and diverse protein subfamilies. I think this also has an impact on the degree to which ancestral sequence recovery can be done with high confidence when using data sets with a high degree of sequence conservation. It seems to me that the limitations of GRASP need to be analyzed/mentioned in this regard.

3 The authors' analysis of CYP2U/CYP2R/CYP2D discusses the differences in structural stability and substrate specificity of the recovered ancestral proteins due to the presence or absence of three characteristic independent insertions. This in itself is a good analysis to see the effect of the presence or absence of indel by ancestral sequence restoration. However, a large part of the effect of the presence/absence of the insertion sequence on structure and function is likely to come from the difference in 3D structure due to the presence/absence of the insertion sequence structure, since the 3D structure of the CYP2 family is known (e.g. PDBid 2F9Q), and homology modeling, etc. can be used to determine the structure of the CYP2 family. It would be helpful to have a comparison of three-dimensional structures to further discuss the effectiveness of the ancestral sequence reconstruction performed in this program.

4 The authors restrict their analysis of data with a high rate of assumed Indels to just However, from a protein engineering perspective, it would be desirable to be able to express and prepare soluble/analyzable ancestor proteins from data with a high percentage of Indel, which can also be described as low sequence similarity. This has advantages, such as increasing the likelihood that enzyme proteins with lower substrate specificity can be made. Real data on such analysis would be very good. Reanalysis of existing data (e.g., reanalysis of ancestral sequence reconstructions from other groups using GRASP) is also acceptable.

Translated with www.DeepL.com/Translator (free version)

Reviewer #3: I asked two expert sub-reviewers, one with an algorithmic bent and the second with a protein and structural bent. Their reports are presented below. Both found the work important and interesting, with remarkably well-made software. However, the former was not able to understand the details of the algorithms, and the latter has several comments, particularly on the length of the indels. Thus, I (we) recommend acceptance pending major revisions, corresponding to the points below. Hope this helps!

First sub-reviewer **********************

From an algorithmic standpoint, the main contribution of this manuscript is the proposal of a new graph-based method to deal with insertions and deletions in ancestral sequence reconstruction (ASR). The new method is named BE (bidirectional-edge encoding) and is implemented in the GRASP software. GRASP is compared against:

- PS (single-site position-specific reconstruction) implemented in PAML, and

- SIC (simple indel encoding) implemented in FastML.

GRASP is capable of handling large trees, and produces reconstructions similar to those of FastML, with the added advantage of presenting them with a graph formalism, which is the natural solution to represent uncertainty in the reconstruction.

I believe that the GRASP software will be a very useful tool for ASR (I played a bit with the tool at http://grasp.scmb.uq.edu.au/ and I'm very positively impressed). However, I have a few comments on the algorithmic side of this work. These are easy to address, although they will require some more writing:

**** (1) ****

I have tried really hard to understand the details of the algorithm at page 32, but I cannot say I understand it. This is important, as this algorithm forms the basis of the new methodology presented here. Here are the aspects that are unclear:

** 1a. About binary matrix E representing the POG, defined at lines 782-790: Is this matrix triangular? That is E(a,b)>0 only for pairs a

** 1b. About equation (1) at line 835:

- The equation would benefit from being decomposed into several equations. It is very hard to read as is.

- When c belongs to J (i.e. c is a leaf, rightmost cases in lines 1 and 3), the added cost is 0, but this sounds wrong to me: this appears to imply that the cost of any edge at a leaf is 0, but this sounds wrong. Only edges in the extant POG E^(j) should receive a cost of 0, while the remaining ones should have a positive cost (1 or infinite?)

- In fact, sigma(j,delta,a,b) for j belonging to J (a leaf) should be explicitly defined somewhere, as it will constitute the base case of the recursion.

** 1c. The sentence "the edge with best score [...] is assigned a score of 1" is confusing (line 831). I guess that the first occurrence of "score" refers to the cost sigma, and the second occurrence refers to E, but I'm not sure.

** 1d. The whole delta = IN, OUT bit sounds very confusing to me. Couldn't the authors just write (a,b) for one direction and (b,a) for the other?

** 1e. The inference for a node k seems to be only based on the inferences for its descendants. This is very unusual, as most ASR algorithms include a bottom-up and a top-down phase, allowing information to flow in all directions within the input tree. Does this mean that to get the marginal reconstruction at a specific node, the tree must be re-rooted in that node? If that's the case, it must be said.

**** (2) ****

The authors do not cite any paper from a whole line of work from the late 2000s about optimization problems for indel history reconstruction, including both maximum parsimony and maximum likelihood formulations. See for example the work in Mathieu Blanchette's group, the papers by Snir and Pachter (2006, 2011), and this PhD thesis for a review: https://escholarship.mcgill.ca/concern/theses/xd07gv166.

I do not consider this to be a major problem, because none of the algorithms developed in those works is likely to be applicable here, due to their poor scalability. Moreover, those works usually assume "phylogenetic correctness" of the ancestral reconstructions, that is, a character that has been deleted cannot be re-inserted back -- a requirement that (interestingly) is not imposed here.

Despite the important differences, I think that acknowledging and discussing some of these works is very important, to set the background.

**** (3) ****

The authors appear to claim that GRASP produces *all* most parsimonious solutions (line 221). However they provide no proof, nor do they provide a clear definition of the parsimony cost function that they wish to optimize. (By the way, for Blanchette's definition of intel parsimony, the problem has been shown to be NP-hard by Chindelevitch et al. 2007, so any exact solution would be very surprising.)

While a formal treatment would probably be beyond the scope of this paper, I think that unproven claims should be avoided. Unless proven otherwise, the algorithms in GRASP are heuristic, and the authors should acknowledge this.

Second sub-reviewer **********************

In line 188, they stated the use of MAFFT to realign the sequences generated by INDELible. I have some experience aligning data coming from alternative splicing with big indels because of the changes in the intron-exon architecture. MAFFT is better than classical aligners, as it tends to create blocks, but it starts to generate errors in the alignments in some conditions, especially with high indel variability and homologous regions (exon duplications). In that context, I have found that ProGraphMSA is a lot better, as it tries to create a phylogenetically sensible gap pattern. I wonder whether such an aligner could better suit this study, but MAFFT is a good choice anyway. All this also makes me wonder how their method performs in the presence of homologous regions inside a protein; is it something challenging for the proposed algorithm?

I have found it strange that they are mainly working with short indels. In line 954, they state that the maximum indel length created by INDELible is 10. Also, in line 633, they said that they had removed sequences that had the longest indel, over 20 amino acids. Therefore, I wonder how the algorithm performs in scenarios with longer indels.

I wonder why they avoid having indels from variations in the exon-intron architecture. They mention in line 637 that they have removed sequences that have a difference in exon number larger than two. Limiting the long indel can also help avoid significant changes in exon composition. Even that, I still wonder whether the CYP2U/2R/2D indel events could be related to alternative splicing or exon-intron architecture changes.

Also, they look to have edited their alignments so much that I wonder whether they have accessed biological insertion and deletions. For example, in line 503, they noted to have trimmed highly gapped columns and used Gblock. I guess that both things can change the number and length of indel events.

In line 423, they talk about the possible structural location of the indels and their flexibility, but, sadly, the work doesn't have a structural analysis of the reconstructed sequences. I was expecting more on that front, but I guess they haven't seen anything interesting. In particular, because they have explicitly selected sequences with structural data available for the DHAD dataset.

In line 578, why haven't they subsampled the tree of all sequences to create the guide trees for the multiple sequence alignments avoiding having tree changes? In any case, I guess they would see the same trend but with more minor variations.

Lastly, did you precisely understand how they filtered the MSA in line 719? The phrase wasn't clear to me.

**Have the authors made all data and (if applicable) computational code underlying the findings in their manuscript fully available?**

Reviewer #1: Yes

Reviewer #2: **No: **As noted in Comments to Authors, the alignments and phylogenetic trees, in addition to resurrected sequences of ancestral proteins seem not to be found in the main text nor supplemental information as the forms which can be accessed by readers.

Reviewer #3: None

PLOS authors have the option to publish the peer review history of their article (what does this mean?). If published, this will include your full peer review and any attached files.

Reviewer #1: No

Reviewer #2: No

Reviewer #3: No
---

## [Decision Letter · Decision Letter 1]

27 Jul 2022

Dear Dr. Boden,

Thank you very much for submitting your manuscript "Engineering indel and substitution variants of diverse and ancient enzymes using Graphical Representation of Ancestral Sequence Predictions (GRASP)" for consideration at PLOS Computational Biology. As with all papers reviewed by the journal, your manuscript was reviewed by members of the editorial board and by several independent reviewers. The reviewers appreciated the attention to an important topic. Based on the reviews, we are likely to accept this manuscript for publication, providing that you modify the manuscript according to the review recommendations.

Sincerely,

Marco Punta

Associate Editor

PLOS Computational Biology

Arne Elofsson

Deputy Editor

PLOS Computational Biology

[LINK]

Reviewer's Responses to Questions

**Comments to the Authors:**

Reviewer #2: The revised version of this paper appears to be well updated from the original manuscript according to the comments of the reviewers, including myself. I have written four comments on the original manuscript. The authors seem to have generally responded appropriately to my comments and revised the manuscript accordingly. On my first comment, I think that authors' response satisfies my comments on the alignments and phylogenetic trees which were not mentioned in the original manuscript. On my fourth comment, I apologize for the grammatical problems in my comment. Despite this, I appreciate that the response by the authors makes it easier to understand how much indel (gap/insertion) exists between sequences in the alignments for inferring ancestral sequences in the paper.

However, I would like to comment further on their responses, which are listed below.

On my second comment (In relation to the above section, it is my understanding that only analyses that do not take into account differences in evolutionary rates between loci when estimating ancestral sequences are valid for GRASP. Note that I have also accessed the web page of the authors' web service. I wonder if such an analysis would be able to recover reasonable amino acid sequences and the presence or absence of indels, especially when using data with low sequence conservation and diverse protein subfamilies. I think this also has an impact on the degree to which ancestral sequence recovery can be done with high confidence when using data sets with a high degree of sequence conservation. It seems to me that the limitations of GRASP need to be analyzed/mentioned in this regard.).

I understand authors' policy on the treatment of heterogeneity of evolutionary rate among sites. The addition of statement on p. 29 seems to make better understandings by readers including me. However, the manuscript states that it is possible to use a phylogenetic tree that assumes heterogeneity in evolutionary rates among sites obtained from the IQ-Tree. Does this mean that the branch lengths assuming heterogeneity of evolutionary rates among sites in the phylogenetic tree given are treated the same as the branch lengths of the phylogenetic tree assuming homogeneity of evolutionary rates among sites for the estimation of ancestral sequences and ancestral indels in GRASP?

On my third comment (The authors' analysis of CYP2U/CYP2R/CYP2D discusses the differences in structural stability and substrate specificity of the recovered ancestral proteins due to the presence or absence of three characteristic independent insertions. This in itself is a good analysis to see the effect of the presence or absence of indel by ancestral sequence restoration. However, a large part of the effect of the presence/absence of the insertion sequence on structure and function is likely to come from the difference in 3D structure due to the presence/absence of the insertion sequence structure, since the 3D structure of the CYP2 family is known (e.g. PDBid 2F9Q), and homology modeling, etc. can be used to determine the structure of the CYP2 family. It would be helpful to have a comparison of three-dimensional structures to further discuss the effectiveness of the ancestral sequence reconstruction performed in this program.).

Authors added the comparison of (predicted) 3D structures of ancestral CYP2Us. I am not familiar on the structure of CYP2Us, it is not easy for me to understand the relationship between AA' loop/D-E loop and indels "LLEE" and "LLSPP". Authors are requested to be re-written the paragraph added at p. 12 of revised manuscript to clarify the relationship between AA' loop/D-E loop and indels "LLEE" and "LLSPP".

Reviewer #3: Again, I used the same two expert sub-reviewers, one with an algorithmic bent and the second with a protein and structural bent. Their reports are presented below. Both acknowledge the authors’ efforts to improve the clarity and address all the referees’ points. However, they still have many comments and concerns. They recommend acceptance with minor (but mandatory) changes, and I agree with this opinion. Hope this helps!

First sub-reviewer **********************

The main comment I had about this manuscript was about the clarity of the new algorithm. Both reviewer 1 and the editor also noted that this had to be improved. Because the authors appear to have put some effort into this, I had another good look into this. Here are two (hopefully constructive) remarks:

A) First, the description of the bi-directional approach still sounds more complicated than needed. If I had to describe it in my own words I would have said something like:

For any given internal node k of the tree, and for any site i, the goal is to infer two things:

- the right neighbor of site i in the sequence at k,

- the left neighbor of site i in the sequence at k.

These can be viewed as two characters that can be separately inferred via standard methods in phylogenetics (max parsimony and likelihood), on the basis of the right/left neighborhood relationships in the observed (extant) sequences.

Barring one detail (see next point), this is probably all that a specialist of ancestral reconstruction will need to understand the bi-directional approach, which is the only methodological novelty here.

B) A point where there remains some ambiguity is the following, and I really believe that this needs to be clarified:

When site i is not present in an extant sequence (ie. at a leaf of the tree), how do we set the characters encoding its left and right neighbor? There are two options:

- we treat them as missing data, which both parsimony and likelihood know how to deal with;

- we treat them as a special character competing with the other possible left/right neighbors. In the revised manuscript the authors say that they have chosen this option. They call this the "no edge" option.

What worries me is that the behavior of the online tool seems to contradict the claims made in the manuscript. All the tests I made at http://grasp.scmb.uq.edu.au/ are consistent with encoding the characters above as missing data, not as "no edge"!

In particular the example in Fig. 6 (introduced in the revised manuscript), allows one to discriminate between the two options above. The online tool launched on this example returns a POG for node N1 where the edges (2,3) and (3,4) have bi-directional support. This is consistent with the "missing data" encoding. However it contradicts the diagrams shown in Fig. 6, where the two edges cited above only have uni-directional support, which is instead consistent with the "no edge" encoding.

(The reason for these observations are technical but easy to understand: site 3 is only observed in one of the 5 input sequences, so its left and right neighbor at node N1 will be assigned the "no edge" option if it's available; however if the neighbors of 3 are treated as missing data, they will be assigned the only observed left and right neighbors, which are 2 and 4 respectively.)

The source of the contradiction above is not due to the fact that the online tool implements max likelihood, while Fig. 6 is based on parsimony: I made sure to provide a tree with constant and very small branch lengths, where likelihood and parsimony are equivalent. Here are the file contents necessary to reproduce this example:

- The tree: ((a:0.001,b:0.001):0.001,c:0.001,(d:0.001,e:0.001):0.001);

- The sequence alignment:

>a

CS-PW

>b

RS-PW

>c

MADPW

>d

MA--W

>e

MA--W

Minor comments

--------------

- The superscripts of E in equations (3) and (4) should be (c), not (j).

- In Fig. 6, E^N2(2,5) should be 0, not 1.

Second sub-reviewer **********************

Thanks to the authors for their detailed responses. I appreciate the work they have done trying to address any concerns and questions — including those questions that I asked out of simple curiosity. Next, you will find my comments on each of the authors’ answers:

(1) Thanks for clarifying the decision behind choosing MAFFT. I have found it interesting that you have found no further improvements in the accuracy of the INDELible MSA reconstructions using ProGraphMSA.

(2) [A] I understand the point made by the authors. Small changes can indeed cause functional changes and have a lesser structural impact. However, it would be better to assess the structural impact of large indels rather than simply assume them to be destabilizing or artifacts. For example, the AA’ loop, a region longer than 20 residues, doesn’t look essential for the overall domain fold. In fact, it is located between the transmembrane region — or included in the membrane — and the PF00067 domain. Note that even single domain proteins can have long embellishment outside the conserved structural core that can come from insertions or deletions events through evolution. In any case, the 20 residue cutoff for CYP2U1 could be acceptable as the fold is relatively rigid. [B] That’s great. I wonder how common those overlaps are in the presented data. It could be great to add the distribution of indel lengths into supplementary materials. That would help the reader understand the indel length range on which you tested the software.

(3) Thank you so much for performing this analysis; the results are interesting, even if they are not decisive. I favor not including ENST00000508453.1. I also agree with the author's decision after looking at some orthologs; 5 to 6 exons sound fine for them. However, filtering everything by the number of exon changes sounds arbitrary. But, I understand that it is not easy to have quality assessments, such as the ones described for CYP2U1, for non-model species.

(4) Thanks for clarifying. Could you split that sentence into two to ease the reader's work? It's not clear how the sequences have been processed. In particular, have they been processed by trimAl?

(5) Thanks for adding the AlphaFold2 structures; it helps to have an idea of the structural location and impact of the indels.

(6) Thanks for the clarification. I think the procedure is ok; I was curious. Now, it is unclear how the sampling was performed to avoid potential bias from the phylogenetic signal. I guess they would never be independent samples given their homology.

(7) Thanks for clarifying this; sadly, it is still unclear to me which variables are used to calculate the entropies. Could you please add the equations? I think that could make it more clear.

**Have the authors made all data and (if applicable) computational code underlying the findings in their manuscript fully available?**

Reviewer #2: Yes

Reviewer #3: Yes

PLOS authors have the option to publish the peer review history of their article (what does this mean?). If published, this will include your full peer review and any attached files.

Reviewer #2: No

Reviewer #3: No

Figure Files:

Data Requirements:

Reproducibility:

References:

---

## [Decision Letter · Decision Letter 2]

27 Sep 2022

Dear Dr. Boden,

Thank you very much for submitting your manuscript "Engineering indel and substitution variants of diverse and ancient enzymes using Graphical Representation of Ancestral Sequence Predictions (GRASP)" for consideration at PLOS Computational Biology. As with all papers reviewed by the journal, your manuscript was reviewed by members of the editorial board and by several independent reviewers. The reviewers appreciated the attention to an important topic. Based on the reviews, we are likely to accept this manuscript for publication, providing that you modify the manuscript according to the review recommendations.

Please note the comment from one reviewer: I would like to comment in the sense that I would like the paper to be easy to read. That is, in some figures, the letters used in the figures are too small and difficult to read. For example, the letters in the graphs c), d), and e) in Figure 3 seem too small to read.

Sincerely,

Arne Elofsson

Section Editor

PLOS Computational Biology

Arne Elofsson

Section Editor

PLOS Computational Biology

[LINK]

Please note the comment from one reviewer: I would like to comment in the sense that I would like the paper to be easy to read. That is, in some figures, the letters used in the figures are too small and difficult to read. For example, the letters in the graphs c), d), and e) in Figure 3 seem too small to read.

Reviewer's Responses to Questions

**Comments to the Authors:**

Reviewer #2: In the revised version of this paper, authors changed parts where I (and other reviewers) commented to be reconsidered. I think that most of them are improved the clearness of this paper. I think this paper is useful for researchers who are doing ancestral sequence resurrection.

I would like to comment in the sense that I would like the paper to be easy to read. That is, in some figures, the letters used in the figures are too small and difficult to read. For example, the letters in the graphs c), d), and e) in Figure 3 seem too small to read.

Reviewer #3: Thanks to the authors for addressing all my comments

**Have the authors made all data and (if applicable) computational code underlying the findings in their manuscript fully available?**

Reviewer #2: Yes

Reviewer #3: Yes

PLOS authors have the option to publish the peer review history of their article (what does this mean?). If published, this will include your full peer review and any attached files.

Reviewer #2: No

Reviewer #3: No

Figure Files:

Data Requirements:

Reproducibility:

References:

---

## [Editor Report · Decision Letter 3]

4 Oct 2022

Dear Dr. Boden,

We are pleased to inform you that your manuscript 'Engineering indel and substitution variants of diverse and ancient enzymes using Graphical Representation of Ancestral Sequence Predictions (GRASP)' has been provisionally accepted for publication in PLOS Computational Biology.

Best regards,

Arne Elofsson

Section Editor

PLOS Computational Biology

Arne Elofsson

Section Editor

PLOS Computational Biology

---

## [Editor Report · Acceptance letter]

14 Oct 2022

PCOMPBIOL-D-22-00025R3 

Engineering indel and substitution variants of diverse and ancient enzymes using Graphical Representation of Ancestral Sequence Predictions (GRASP)

Dear Dr Bodén,

I am pleased to inform you that your manuscript has been formally accepted for publication in PLOS Computational Biology. Your manuscript is now with our production department and you will be notified of the publication date in due course.

With kind regards,

Zsanett Szabo
